complexity

complex network, maritime transportation, international trade

**Author for correspondence:**
Naoki Masuda
e-mail: naokimas@buffalo.edu

# Estimating international trade status of countries from global liner shipping networks

Mengqiao Xu[1], Qian Pan[1], Haoxiang Xia[1]
and Naoki Masuda[1,2,3]

[1]School of Economics and Management, Dalian University of Technology, No. 2 Linggong Road, Ganjingzi District, Dalian City, Liaoning Province 116024, People's Republic of China
[2]Department of Mathematics, University at Buffalo, Buffalo, NY 14260-2900, USA
[3]Computational and Data-Enabled Science and Engineering Program, University at Buffalo, State University of New York, Buffalo, NY 14260-5030, USA

 MX, 0000-0001-8234-7910; QP, 0000-0002-6365-2781;
HX, 0000-0001-5737-1214; NM, 0000-0003-1567-801X

Maritime shipping is a backbone of international trade and, thus, the world economy. Cargo-loaded vessels travel from one country's port to another via an underlying port-to-port transport network, contributing to international trade values of countries en route. We hypothesize that ports that involve trans-shipment activities serve as a third-party broker to mediate trade between two foreign countries and contribute to the corresponding country's status in international trade. We test this hypothesis using a port-level dataset of global liner shipping services. We propose two indices that quantify the importance of countries in the global liner shipping network and show that they explain a large amount of variation in individual countries' international trade values and related measures. These results support a long-standing view in maritime economics, which has yet to be directly tested, that countries that are strongly integrated into the global maritime transportation network have enhanced access to global markets and trade opportunities.

## 1. Introduction

International trade is important for the economic growth of countries [1–3]. Maritime countries (i.e. countries that are not landlocked) altogether account for approximately 92% of the total value of international trade, and more than 80% of the commodity cargo worldwide (in terms of volume) are transported by ships [4]. As such, maritime shipping is a backbone of international trade and thus the world economy [5–7].

Therefore, data on maritime shipping and ports may provide useful information on international trade and its growth. First, the World Bank has been financing more than 360 port and waterway

construction projects in 104 countries and regions since the 1950s, with a total investment of more than US$ 21.4 billion [8]. In fact, the growth in trade between a pair of countries was found to be correlated with how early the two countries first adopted port containerization (i.e. processing of container cargos transported by container vessels) [9]. This result suggests that port containerization is a variable that may be closely related to world trade growth. Second, since its inception in 2004, the United Nations Conference on Trade and Development's (UNCTAD) liner shipping connectivity index (LSCI) has been an official indicator of maritime transport in the UNCTAD statistics [10]. The LSCI is computed for individual economies (we simply call them countries) and aims to quantify the extent to which each country is integrated into the global liner shipping network (GLSN). Note that liner shipping, i.e. the service of transporting goods primarily by ocean-going container ships that follow regular routes on pre-fixed schedules, accounts for more than 70% of the cargo value transported by sea [4]. The liner shipping bilateral connectivity index (LSBCI), which is a variant of the LSCI and computed for a pair of countries rather than single countries, quantifies the extent to which a country pair is integrated into the GLSN [10]. The LSBCI was found to be correlated with South Africa's bilateral trade values [11]. Third, the Baltic Dry Index (BDI) is an indicator of average global freight rates for transporting major raw materials (i.e. coal, iron ore, crude oil and grain) [12]. The BDI was correlated with the prices of stock, currency and commodities futures markets over 3–5 years, thus it is promising as a signal to predict short-term growths of total international trade [13]. Fourth, shipping cost was recently found to negatively impact trade development even for landlocked developing countries [14].

The aim of the present study is to evaluate the extent to which the information on the GLSN connectivity (i.e. which ports connect to which ports) helps us to estimate countries' international trade status. A seminal study constructed networks of ports based on itineraries of cargo ships to reveal their structural properties [15]. Shipping networks have been shown to be useful in understanding trading communities [15–18], port performance ranking [16,19], vulnerability of the global liner shipping system [20–22], the spread of marine bioinvasion [23–26] and maritime traffic monitoring [27]. The information provided by such concrete shipping networks is considered to be complementary to that provided by the existing measures such as the degree of containerization and LSCI. These existing measures quantify how much individual countries or ports are integrated into international trade and the global economy. However, they do not tell how countries or ports are specifically connected to each other.

A long-standing premise in maritime economics is that countries that are strongly integrated into the global maritime transportation network have enhanced access to global markets and trade opportunities [9]. To operationally test this claim, we hypothesize that the role of a port or country as broker to mediate liner shipping between different countries is correlated with the importance of the port or country in international trade. This hypothesis is consistent with both liner shipping industry practices and network theory. In liner shipping, a broker role may reflect the potential of a port/country to be a trans-shipment hub that facilitates container cargo transportation between other ports/countries. In fact, because the trans-shipment of containers has been a fastest-growing segment of the container port market, container ports are fiercely competing for becoming trans-shipment hubs [28,29]. In network analysis, various centrality measures for nodes quantify the importance or role of nodes under the premise that the node's position impacts opportunities and constraints that it encounters [30,31]. In particular, the role as broker is often quantified by the betweenness centrality [32] or more succinctly by the degree (i.e. the number of edges that a node has). However, these or other centrality measures largely use only the data about the network structure and neglect metadata, such as the nationality of the ports or the individual service routes. Such centrality measures based only on the network structure may be poor indicators of countries' statuses in international trade and global economy.

In the present study, we analyse port-level GLSNs, which we derived from a record of liner shipping services. We propose two GLSN-based indices for individual countries that quantify each country's role as broker in international maritime transport. Although the two indices are analogous to the node's degree and betweenness, the new indices use the information on ports' nationalities and on the individual service routes. Then, we show that the proposed indices account for the country's international trade value fairly well. In particular, their performance, either alone or in combination, is found to perform better than previously established liner connectivity metrics such as the LSCI.

## 2. Methods

### 2.1. Data

The dataset was provided by Alphaliner [33] at a given point of time (i.e. April 2015) and was also used in our previous work [17,34]. The Alphaliner database extensively covers the fleets of regular service

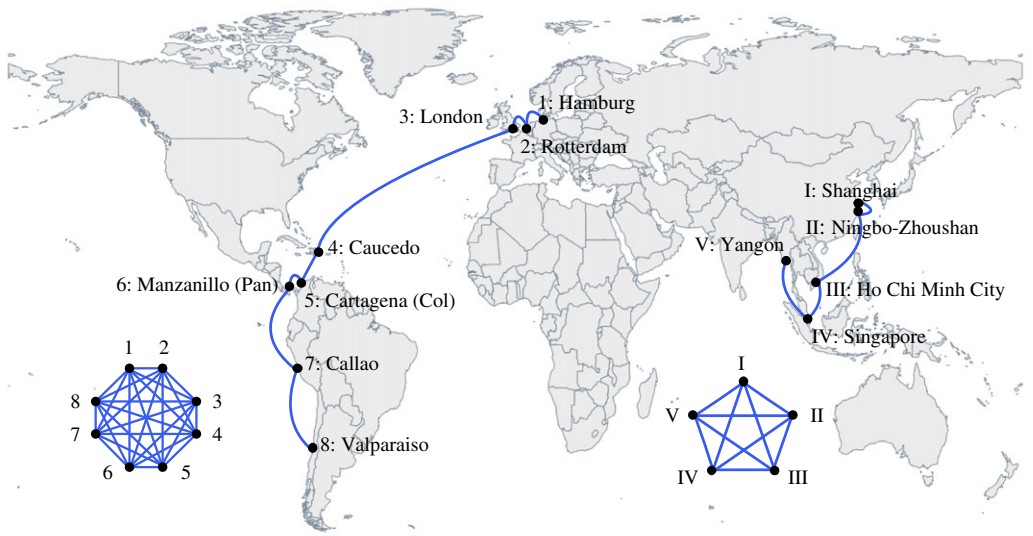

**Figure 1.** Construction of the unweighted GLSN. We show, with two examples of service routes, how each service route induces a clique such that any pair of ports in the same service route is adjacent to each other.

routes worldwide, including all the service routes of the world's top 100 liner shipping companies in terms of liner shipping capacity (measured in Twenty-foot Equivalent Unit (TEU)). Note that the top 100 companies altogether account for approximately 92% of world's total liner shipping capacity [35]. The vessel types included in the dataset are full-container vessels and multi-purpose vessels. We use the information on 1316 international liner shipping service routes (service route for short), all of which were deployed with full-container vessels. We exclude service routes with multi-purpose vessels because service routes with full-container vessels are the most common in liner shipping practice [36]. Therefore, the cargo type is containerized cargo. The information on the cargo size for each ship is not available. The data on each service route include the ports of origin and destination, midway load/unloading ports, and the order in which the ports are visited, but not refuelling ports.

We focus on international service routes (international routes for short). Specifically, a service route is international if it includes ports of different countries. The 1316 international routes with full-container vessels contain 777 ports located in 178 countries. It should be noted that we use the term country interchangeably with the term economy. Therefore, a country does not imply political independence but refers to any territory for which authorities report separate social or economic statistics.

Among the 178 countries, the trade value, i.e. the sum of the merchandize export and import value (in current US$), and the LSCI for 157 countries, both in the year 2015, were available in the World Bank database [37] and in the UNCTAD database [10], respectively. For these 157 countries, we collected the GDP statistics (in current US$) of 151 countries from the World Bank database and that of the other six countries (i.e. Cayman Islands, Eritrea, New Caledonia, French Polynesia, Syrian Arab Republic, and Venezuela (Bolivarian Republic of)) from the UNdata database [38]. They altogether account for approximately 92% of the world's total trade value.

## 2.2. Construction of the GLSN

On each service route, container ships call at a sequence of ports with a fixed service schedule. In general, a single ship can transport cargo between any two ports on a service route. Therefore, we constructed an unweighted GLSN, in which a node represents a port, as follows. First, each service route forms a clique such that any pair of ports in the same service route is connected to each other, as shown in figure 1. Then, by overlapping all the cliques derived from the individual service routes and ignoring the edge weight, we obtained an unweighted GLSN that consists of 777 nodes and 12 000 edges.

We also constructed six weighted GLSNs that have the same network structure as that of the unweighted GLSN as follows. Consider a service route that contains $n$ ports and is deployed by (possibly multiple) world shipping companies with a pre-fixed total traffic capacity (measured in TEU), denoted by $C$. The Alphaliner dataset provides the $C$ value for each service route. We assigned to any pair of ports belonging to this route the same edge weight that is equal to either 1, $1/(n-1)$,

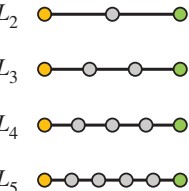

**Figure 2.** Schematic illustration of the valid shortest paths. Valid shortest paths are the shortest paths between two end ports belonging to two different countries (shown in yellow and green) that exclusively contain ports of other countries as intermediate ports (shown in grey). The figure shows valid shortest paths of different lengths.

$1/[n(n-1)/2]$, $C$, $C/(n-1)$, or $C/[n(n-1)/2]$, in terms of the TEU. The first three edge-weighting methods neglect the traffic capacity of each route, $C$, whereas the last three methods use it. If each port accounts for a total traffic capacity of $C$, which is equally divided by its potential partner ports, then each edge receives an edge weight of $C/(n-1)$. Alternatively, a ship may transport cargos between any pair of ports in a relatively even manner. Therefore, the normalization factor $C/[n(n-1)/2]$ implies that $C$ is equally divided by all the possible $n(n-1)/2$ pairs of ports. In our previous work, we adopted $C/(n-1)$ as the edge weight to analyse a similarly constructed GLSN [17]. For each of the six edge-weighting schemes, we calculated the edge weight for a given pair of ports as the summation of the edge weight over all the service routes to which both ports belong.

## 2.3. Explanatory variables

In descriptive analysis and the multivariate linear regressions, we used the following four GLSN-related explanatory variables, each of which we measured for the individual countries. In particular, we examined the power of these four variables in explaining countries' trade values and their growth.

### 2.3.1. GLSN connectivity

The GLSN connectivity of country $i$ aims at capturing the extent to which a country is connected with the rest of the world in the GLSN. We define the GLSN connectivity as the sum of the edge weight over the edges between any port of country $i$ and any foreign port. This definition applies to both unweighted and weighted GLSNs.

For a given unweighted or weighted GLSN, we also considered the normalized GLSN connectivity of a country. We define the normalized GLSN connectivity of country $i$ by dividing the original GLSN connectivity by the number of ports in country $i$.

### 2.3.2. GLSN betweenness

We introduce the so-called GLSN betweenness, which is a variant of betweenness centrality. Consider a pair of ports $s$ and $t$ belonging to different countries (yellow and green nodes in figure 2) and a shortest path connecting them in the GLSN. We call the shortest path valid when its length is less than or equal to $L_{\max}$ and each port on the shortest path except $s$ and $t$ (grey nodes in figure 2) belongs to a country different from the countries of $s$ and $t$. We treat $L_{\max}$ as a parameter and set $L_{\max} = 2$, 3, 4 or 5. We did not consider larger $L_{\max}$ because the longest shortest path in the GLSN is of length 5. A longer valid shortest path may represent a more complicated transportation scenario such as more times of trans-shipment. We hypothesize that ports located on the valid shortest path between ports $s$ and $t$, excluding $s$ and $t$, are crucial for international trade because they influence the trans-shipment and thus the accessibility of cargo transportation between the two countries represented by ports $s$ and $t$.

The GLSN betweenness of country $i$ is defined to be the fraction of the valid shortest paths when one varies $s$ or $t$, in which any port of country $i$ appears between $s$ and $t$ (grey nodes in figure 2). If there exist more than one valid shortest paths between $s$ and $t$, then each valid shortest path is given equal weight, i.e. $1/n^{st}$, where $n^{st}$ is the number of valid shortest paths between $s$ and $t$. In this manner, the sum of the weight over all the valid shortest paths between $s$ and $t$ is equal to 1. The GLSN betweenness of a country $i$, denoted by $Gb_i$, is given by

$$Gb_i = \sum_{s<t} \left( \frac{g_i^{st}}{n^{st}} \right), \tag{2.1}$$

where $g_i^{st}$ is the number of valid shortest paths between $s$ and $t$ that include at least one port of country $i$ as an intermediate port.

### 2.3.3. Freeman betweenness

The betweenness centrality of the $i$th node, denoted by $b_i$, is defined as

$$b_i = \sum_{s<t} \left( \frac{\sigma_i^{st}}{\rho^{st}} \right), \tag{2.2}$$

where $\rho^{st}$ is the number of shortest paths between nodes $s$ and $t$, and $\sigma_i^{st}$ is the number of shortest paths between $s$ and $t$ passing through $i$ [39]. We first calculated each port's betweenness centrality in the unweighted GLSN. Then, as we did for the GLSN connectivity, we defined the betweenness centrality of country $i$ as the sum of the betweenness centrality of the ports belonging to country $i$. We define the normalized betweenness centrality as the average of the port's betweenness centrality over the ports in country $i$. To distinguish these betweenness measures from the GLSN betweenness, we refer to the former as Freeman betweenness and normalized Freeman betweenness.

### 2.3.4. LSCI

The LSCI, originally developed in 2004 and improved in 2019 by UNCTAD, is an indicator for the extent of countries' integration into the existing GLSN [10]. It is calculated based on the following six components: (i) the number of scheduled ship calls per week in the country; (ii) annual capacity in terms of TEU, which means the total container-carrying capacity that the world's shipping companies offer to the country; (iii) the number of regular liner shipping services visiting the country; (iv) the number of liner shipping companies that provide services from and to the country; (v) the largest of the average vessel size among all the scheduled services involving the country, where the average vessel size for a scheduled service is defined as the average size of the vessels deployed on the scheduled service in terms of the TEU; and (vi) the number of other countries that are connected to the country through single liner shipping services.

## 2.4. Statistical models

We adopted multivariate linear regressions to explain the trade value of countries. To check the collinearity between independent variables to justify the use of the multivariate linear regression, we measured the variance inflation factor (VIF) for each independent variable [40,41]. The VIF is the reciprocal of the fraction of the variance of the independent variable that is not explained by linear combinations of the other independent variables. Large values of VIFs indicate that the associated regression coefficients are poorly estimated due to collinearity. In many empirical studies, VIFs smaller than 5 are preferred for the multivariate linear regression to be valid [42]. Therefore, we use the same criterion.

We selected the best combination of explanatory variables in multivariate linear regression using Akaike's information criterion (AIC). In the case of least-squares regression analyses as adopted by the present paper, AIC is calculated as

$$\text{AIC} = N \times \ln \left( \frac{\text{RSS}}{N} \right) + 2K, \tag{2.3}$$

where $N$ is the number of observations, RSS is the residual sum of squares of the model, and $K$ is the number of fitted parameters including the intercept.

## 2.5. Gravity model of bilateral trade flows

We consider the following standard gravity model that explains the bilateral trade flows between countries [43]:

$$\ln(\text{BTV}_{ij}) = \beta_0 + \beta_1 \times \ln(\text{GDP}_i \times \text{GDP}_j) + \beta_2 \times \ln(d_{ij}) + \varepsilon_{ij}, \tag{2.4}$$

where $\text{BTV}_{ij}$ is the current US dollar value of the trade between countries $i$ and $j$, $\text{GDP}_i$ is the US dollar value of the nominal GDP for country $i$, $d_{ij}$ is the geographical distance between the economic centre of $i$ and that of $j$, and $\varepsilon_{ij}$ is an error term. We regard a country's capital as its economic centre.

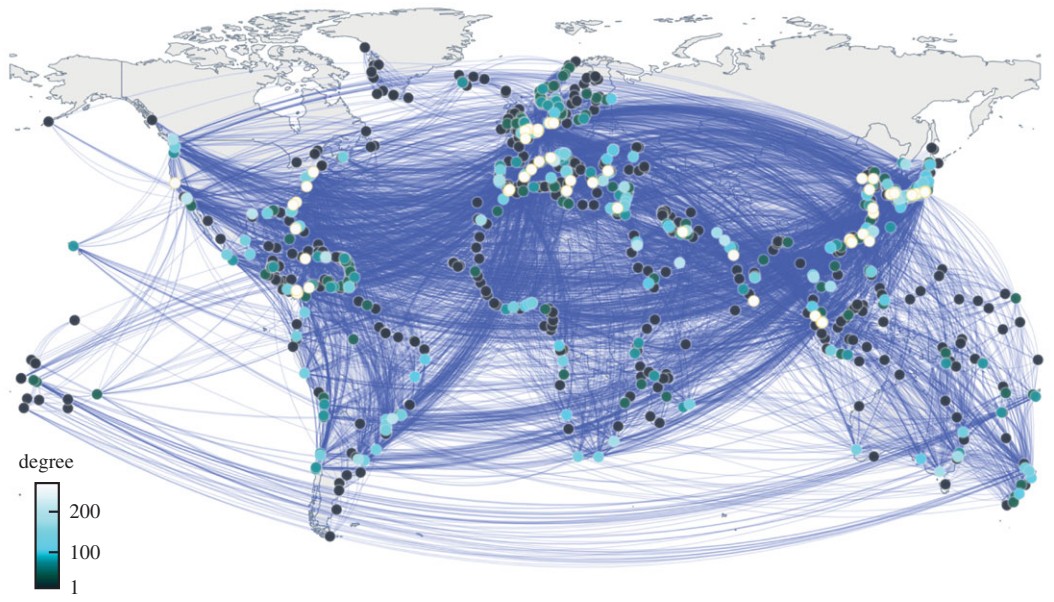

**Figure 3.** Unweighted GLSN. Ports with a large degree are represented by circles in a bright colour.

# 3. Results

We constructed one unweighted and six weighted undirected GLSNs (in terms of traffic capacity measured in TEU) from a dataset of 1316 international liner shipping service routes in 2015. The unweighted GLSN is shown in figure 3.

## 3.1. GLSN connectivity and GLSN betweenness are strongly associated with the country's trade value

The Pearson correlation coefficient between the trade value and each of the explanatory variables based on the 157 countries is shown in table 1. Many explanatory variables based on the GLSN are strongly correlated with the trade value, often with a correlation coefficient larger than 0.85. Given the results shown in table 1, we selected the explanatory variables to be used in the multivariate regression analysis in the following manner. First, we keep the (unnormalized) unweighted GLSN connectivity and drop the six (unnormalized) weighted GLSN connectivity measures, because the former is nearly the top performer in terms of the correlation with the trade value among the different edge-weighting schemes. Second, we drop the normalized GLSN connectivity measures (i.e. the GLSN connectivity divided by the number of ports in a given country), both unweighted and weighted ones, because they are much less correlated with the trade value than the unnormalized counterparts are. Third, we keep the GLSN betweenness with valid shortest paths whose length is less than or equal to 2 (denoted by $L_{max} = 2$) and drop it with larger $L_{max}$. This is because the GLSN betweenness with $L_{max} = 2$ is already reasonably strongly correlated with the trade value and because the existence of valid shortest paths of longer length depends on the existence of valid shortest paths of short length. Fourth, we keep the Freeman betweenness and drop the normalized Freeman betweenness (i.e. the Freeman betweenness divided by the number of ports in a given country), because the former is much more strongly correlated with the trade value than the latter is. Fifth, we keep LSCI because it is an UNCTAD's official indicator, is the only one that we use and does not explicitly depend on our GLSN, and is reasonably strongly correlated with the trade value. Therefore, the LSCI serves as a benchmark indicator. The Gc, Gb, Fb and LSCI values of the 157 countries are shown in electronic supplementary material, figure S1.

## 3.2. Estimating individual country's trade value by multivariate linear regression

We carried out multivariate linear regression, aiming to explain the trade value of different countries by a linear combination of the four explanatory variables identified in the previous section, i.e. GLSN

**Table 1.** Pearson correlation coefficient between the trade value and each explanatory variable when 157 countries are considered. We denote by $r$ the Pearson correlation coefficient. $^{**}p$-value $< 0.001$, $^{*}p$-value $< 0.01$, $^{+}p$-value $< 0.05$.

| variable | | $r$ | variable | | $r$ | variable | | $r$ |
|---|---|---|---|---|---|---|---|---|
| GLSN connectivity | edge weight | — | normalized GLSN connectivity | edge weight | — | GLSN betweenness | $L_{max}$ | — |
| | None | 0.877** | | none | 0.289** | | 2 | 0.793** |
| | 1 | 0.851** | | 1 | 0.248* | | 3 | 0.851** |
| | $1/(n-1)$ | 0.834** | | $1/(n-1)$ | 0.218* | | 4 | 0.852** |
| | $1/[n(n-1)/2]$ | 0.790** | | $1/[n(n-1)/2]$ | 0.175+ | | 5 | 0.852** |
| | $C$ | 0.871** | | $C$ | 0.272** | Freeman betweenness | | 0.889** |
| | $C/(n-1)$ | 0.874** | | $C/(n-1)$ | 0.268** | normalized Freeman betweenness | | 0.308** |
| | $C/[n(n-1)/2]$ | 0.878** | | $C/[n(n-1)/2]$ | 0.259* | LSCI | | 0.749** |

**Table 2.** Results for multivariate linear regressions when the dependent variable is the country's trade value and 157 countries are considered. Gc: GLSN connectivity, Gb: GLSN betweenness, Fb: Freeman betweenness, L: LSCI. Adjusted $R^2$, i.e. adjusted coefficient of determination, measures the proportion of variance explained by the regression and is equal to $1 - [(1-R^2) \times (N-1)/(N-K-1)]$, where $R^2$ is the coefficient of determination, $N$ is the number of observations, and $K$ is the number of explanatory variables. $^{**}p$-value $< 0.001$.

| explanatory variable | adjusted $R^2$ | AIC | Max VIF |
|---|---|---|---|
| Gc | 0.768** | −227.44 | 1.00 |
| Gb | 0.626** | −152.37 | 1.00 |
| Fb | 0.788** | −241.80 | 1.00 |
| L | 0.558** | −126.08 | 1.00 |
| Gc, Gb | 0.812** | −259.51 | 2.22 |
| Gc, Fb | 0.838** | −282.51 | 3.78 |
| Gc, L | 0.772** | −229.13 | 2.83 |
| Gb, Fb | 0.811** | −258.23 | 9.71 |
| Gb, L | 0.658** | −165.42 | 2.87 |
| Fb, L | 0.789** | −241.10 | 2.98 |
| Gc, Gb, Fb | 0.841** | −284.40 | 20.27 |
| Gc, Gb, L | 0.813** | −259.31 | 3.93 |
| Gc, Fb, L | 0.838** | −282.05 | 4.55 |
| Gb, Fb, L | 0.815** | −260.62 | 10.46 |
| Gc, Gb, Fb, L | 0.840** | −282.64 | 20.95 |

connectivity, GLSN betweenness, Freeman betweenness, and LSCI. We ran regression on each of the 15 combinations of the explanatory variables and measured the AIC, adjusted $R^2$, and maximum VIF.

The results of the regressions are shown in table 2. Regression models with the maximum VIF value larger than 5 suffer from collinearity between the explanatory variables in general and therefore should be excluded [40]. It should be noted that there exists severe collinearity between the GLSN betweenness and the Freeman betweenness, with a Pearson correlation coefficient of 0.947; all the models containing both of them were excluded by the VIF criterion. Eleven out of the

15 combinations of the explanatory variables had the maximum VIF value smaller than 5. Among these remaining model configurations, the two-variable model with the GLSN connectivity and Freeman betweenness is the best in terms of the AIC and explains 83.8% of the countries' trade value variance in terms of the adjusted $R^2$.

Next, we investigated the extent to which a country's trade value is explained by local structure of the GLSN, i.e. the country's ports and their neighbouring foreign ports. Therefore, we removed the Freeman betweenness, which requires information about the global structure of the network. In this case, the two-variable model containing the GLSN connectivity and GLSN betweenness performed the best in terms of the AIC and explained 81.2% of the trade value variance. Note that the combination of these two explanatory variables was also selected when one imposed a stricter threshold on the VIF equal to 3.3 [44] and did not exclude the Freeman betweenness. Also note that LSCI is not included in either selected model, although it has long been a prevalent measure of country's integration into the GLSN and the access to world markets [10,45].

To examine generalizability of these results, we then replaced the dependent variable by the export value, import value, and net export value (i.e. export minus import, which is a compound of GDP), which are commonly used trade statistics representing a country's macroeconomic status. First, the export value was strongly correlated with each explanatory variable (GLSN connectivity: 0.868, $p < 10^{-4}$; GLSN betweenness: 0.835, $p < 10^{-4}$; Freeman betweenness: 0.897, $p < 10^{-4}$; LSCI: 0.755, $p < 10^{-4}$). When the export value was the dependent variable, the best linear regression model remained the two-variable model composed of the GLSN connectivity and the Freeman betweenness, and it explained 83.9% of the variance in the export value (electronic supplementary material, table S1). The best model when the Freeman betweenness was excluded contained the GLSN connectivity, GLSN betweenness, and LSCI, explaining 83.6% of the variance. However, in this three-variable model, the LSCI did not have significant explanatory power, whereas the GLSN connectivity and GLSN betweenness did (electronic supplementary material, table S2).

Second, the import value was also strongly correlated with each explanatory variable (GLSN connectivity: 0.863, $p < 10^{-4}$; GLSN betweenness: 0.729, $p < 10^{-4}$; Freeman betweenness: 0.856, $p < 10^{-4}$; LSCI: 0.722, $p < 10^{-4}$). When the import value was the dependent variable, the best model was again composed of the GLSN connectivity and the Freeman betweenness, explaining 79.3% of the variance (electronic supplementary material, table S1). When Freeman betweenness was excluded, the best model contained the GLSN connectivity and GLSN betweenness and explained 75.9% of the variance.

Third, the net export value was only significantly correlated with the GLSN betweenness, and the correlation was not large (GLSN connectivity: −0.005, $p = 0.951$; GLSN betweenness: 0.301, $p < 0.001$; Freeman betweenness: 0.101, $p = 0.208$; LSCI: 0.081, $p = 0.315$). Consistent with this result, the best regression model, which contained the GLSN connectivity, GLSN betweenness and LSCI, only accounted for 20.4% of the variance (electronic supplementary material, table S1). All the four explanatory variables ignore the information on the directionality of inter-port connections and that of international trades. Therefore, they are unable to distinguish the export and import value of a country, which the calculation of the net export of a country requires. We consider that this is a main reason why our regression models only marginally explain the net export value.

## 3.3. Estimating GDP

The gross domestic product (GDP) is a primary indicator used for assessing the size of a country's economy. The GDP represents the total value of all goods and services produced over a specific time period. Previous studies on transport economics have shown that high performance of maritime transport logistics contributes to the economic growth of a country [46,47]. Therefore, in this section, we examine the extent to which the four explanatory variables, which are not direct derivatives of the GDP, are associated with the GDP. We used the GDP at purchaser's prices collected from the World Bank.

The GDP was significantly correlated with each of the four explanatory variables (GLSN connectivity: 0.822, $p < 10^{-4}$; GLSN betweenness: 0.557, $p < 10^{-4}$; Freeman betweenness: 0.741, $p < 10^{-4}$; LSCI: 0.541, $p < 10^{-4}$). We then ran multivariate linear regressions on the combinations of the explanatory variables. The best model in terms of the AIC was composed of the GLSN connectivity, Freeman betweenness and LSCI (electronic supplementary material, table S1), and explained 74.0% of the variance of the GDP. When the Freeman betweenness was removed, the best model was composed of the GLSN connectivity and LSCI, and explained 71.3% of the variance. In contrast to our previous regression results, the LSCI remained in the selected models when the GDP was the dependent variable. However, the LSCI alone explained merely 28.8% of the variance (electronic supplementary

material, table S1). Furthermore, the contribution of the LSCI to the GDP was negative (electronic supplementary material, table S2), which is difficult to interpret. Therefore, these results suggest that the information about a country's position in the GLSN, as measured by the GLSN connectivity and Freeman betweenness rather than the LSCI, considerably contributes to explaining the GDP.

## 3.4. Validation using the data in 2017

We ran the multivariate linear regression of trade value on the four explanatory variables using the GLSN data and trade value data in 2017. It should be noted that the GLSN data in 2017 were the most recent available to us. The results were qualitatively the same as those for the 2015 data (electronic supplementary material, table S4). Specifically, the best model in terms of the AIC remained the one composed of GLSN connectivity and the Freeman betweenness, and the best model without Freeman betweenness remained the one composed of the GLSN connectivity and the GLSN betweenness. Moreover, the estimated coefficients of the 2017 models (electronic supplementary material, table S2) were of similar magnitudes to those of the 2015 models.

## 3.5. Estimating changes in the country's trade value over 3 years

Next, we investigated whether the country's position in the GLSN predicts changes in the trade value over time. We carried out multivariate linear regression to explain the change in the trade value between the years 2015 and 2018 in terms of the four explanatory variables in 2015. We also included the trade value in 2015 (denoted by Tv2015) as an explanatory variable because we expect that the increment/decrement in the trade value in 3 years tends to be large if the trade value itself is large. We decided to use a 3-year interval because maritime shipping markets usually experience short *Kitchin* economic cycles of a 3–4-year period in shipping demand and supply adjustments [48].

Among multivariate linear regression models with all the 31 possible combinations of the five explanatory variables, the best model in terms of the AIC that are free of collinearity (i.e. max VIF < 5) was composed of Tv2015 and the GLSN betweenness (in 2015; denoted by Gb2015) (electronic supplementary material, table S5). The contribution of Tv2015 was by far the largest and explained most of the variance in this and all other models that included Tv2015. However, the contribution of the GLSN betweenness was also significant in the best model (i.e. (Tv2018 − Tv2015) = $\beta_{Tv}$ × Tv2015 + $\beta_{Gb}$ × Gb2015 + intercept, where $\beta_{Tv}$ = 0.789 with the 95% confidence interval (CI) = [0.718, 0.860], $\beta_{Gb}$ = 0.209 with CI = [0.138, 0.280]; an adjusted $R^2$ = 0.927; max VIF = 2.684; we standardized all the explanatory variables). These two variables, but not any other, were also significant in the model composed of all the five explanatory variables ((Tv2018 − TV2015) = $\beta_{Tv}$ × Tv2015 + $\beta_{Gc}$ × Gc2015 + $\beta_{Gb}$ × Gb2015 + $\beta_{Fb}$ × Fb2015 + $\beta_{L}$ × L2015 + intercept, where $\beta_{Tv}$ = 0.843 with CI = [0.734, 0.952], $\beta_{Gc}$ = 0.019 with CI = [−0.095, 0.134], $\beta_{Gb}$ = 0.326 with CI = [0.163, 0.490], $\beta_{Fb}$ = −0.206 with CI = [−0.419, 0.008], $\beta_{L}$ = 0.024 with CI = [−0.062, 0.111]; an adjusted $R^2$ = 0.927; max VIF = 24.564). These results further support the capability of the GLSN betweenness in explaining the trade value of the country.

## 3.6. Comparison with the gravity model

For international trade, the gravity model [43] has long been successful in explaining empirical trade flows between countries and also gained microeconomic foundations [49–51]. Therefore, we compare the explanatory power of the multivariate linear regression with that of the gravity model.

Among the 157 countries analysed in the previous sections, here we analysed 144 countries that we selected as follows. For each country $i$ among the 157 countries, we calculated $\sum_{j \neq i; BTV_{ij}^{emp} > 0} BTV_{ij}^{emp}$, where $BTV_{ij}^{emp}$ is reported in the UN Comtrade database [52]. Note that $j$ does not have to be a country in our GLSN. If and only if this sum is more than 90% of the country's total trade value as reported either by the World Bank or by the UN Comtrade, we used country $i$. In this situation, we consider that the bilateral trade values, which the gravity model is based on, are sufficiently representative of the total trade value.

We applied the gravity model (equation (2.4)), where $i$ is one of the 144 countries and $j$ is any trading partner of $i$, i.e. a country having $BTV_{ij}^{emp} > 0$. The model yielded an adjusted $R^2$ value of 0.680, where the qualified $(i, j)$ pairs were regarded as samples. We then estimated the trade value for country $i$ as $\sum_{j \neq i} BTV_{ij}$, where $BTV_{ij}$ is the value estimated for bilateral trade value between countries $i$ and $j$. The Pearson correlation coefficient between the empirical and estimated trade value of countries was equal to 0.840, resulting in an adjusted $R^2$ value of 0.704.

**Table 3.** Multivariate regression results for gravity models in which the bilateral trade value of country pairs is the dependent variable, and the GLSN betweenness and LSBCI are additional explanatory variables. We considered 1773 country pairs, for which the two countries are directly connected in the GLSN (i.e. there exists at least one connection between the ports of two countries), the two countries' GDP values are available, the LSBCI value between the two countries is available, and the two countries' Gb values are positive. $^{**}p$-value $< 0.001$.

| explanatory variable | adjusted $R^2$ | AIC | Max VIF |
|---|---|---|---|
| $\ln(GDP_i \times GDP_j)$, $\ln(d_{ij})$ | $0.778^{**}$ | 1175.23 | 1.23 |
| $\ln(GDP_i \times GDP_j)$, $\ln(d_{ij})$, $\ln(LSBCI_{ij})$ | $0.785^{**}$ | 1120.65 | 1.94 |
| $\ln(GDP_i \times GDP_j)$, $\ln(d_{ij})$, $\ln(Gb_i \times Gb_j)$ | $0.787^{**}$ | 1102.86 | 2.23 |
| $\ln(GDP_i \times GDP_j)$, $\ln(d_{ij})$, $\ln(LSBCI_{ij})$, $\ln(Gb_i \times Gb_j)$ | $0.787^{**}$ | 1099.83 | 4.24 |

To compare the performance between the gravity model and our GLSN-based linear regression, we re-ran the multivariate linear regression for the subset of the data composed of the 144 countries. The best model in terms of the AIC when all the four explanatory variables were used was the one based on the GLSN connectivity and the Freeman betweenness (adjusted $R^2 = 0.838$; electronic supplementary material, table S3). When the Freeman betweenness was excluded, the selected model was the two-variable one with the GLSN connectivity and GLSN betweenness (adjusted $R^2 = 0.811$; electronic supplementary material, table S3). These two models perform considerably better than the gravity model in terms of the adjusted $R^2$ value.

Furthermore, we assessed whether adding GLSN indicators to the gravity model improves the performance of the gravity model in estimating the bilateral trade value. This approach is in line with the work of Fugazza & Hoffmann [53], which shows that the liner shipping connectivity between two countries as measured by LSBCI helps explain the bilateral trade value between them within the framework of the gravity model. Here, we used $\ln(Gb_i \times Gb_j)$ and $\ln(LSBCI_{ij})$ as two additional explanatory variables to extend the original gravity model. As shown in table 3, in terms of the adjusted $R^2$ value, the model comprising $\ln(GDP_i \times GDP_j)$, $\ln(d_{ij})$ and $\ln(Gb_i \times Gb_j)$ performs as efficiently as the model comprising $\ln(GDP_i \times GDP_j)$, $\ln(d_{ij})$ and $\ln(LSBCI_{ij})$. Both models were free of collinearity (i.e. max VIF $< 5$) and slightly better than the original gravity model, which comprises $\ln(GDP_i \times GDP_j)$ and $\ln(d_{ij})$. The three explanatory variables are all significant ($p < 0.001$) for both extended gravity models. These results suggest that the GLSN betweenness is almost as good as the LSBCI in improving the performance of the gravity model in estimating the bilateral trade value. Additionally, the extended gravity model that used the GLSN connectivity instead of the GLSN betweenness performed similarly (electronic supplementary material, table S6).

## 4. Discussion

The GLSN originates from multiple decisions on service network design made by individual shipping companies worldwide, which primarily aim to maximize profits in a decentralized manner. We hypothesized that the structure of the GLSN is an exogenous transportation factor that not only physically supports but also influences international trade values. Based on a comprehensive port-level GLSN dataset, we constructed GLSNs and proposed network indicators to quantify individual countries' positions in the GLSN. We showed that a country's position in the GLSN was a strong signature of the country's international trade value. In particular, we proposed the GLSN connectivity and GLSN betweenness indices, which one can calculate from local information about the network around the ports of the focal country. The two indices explained the trade value fairly well. The GLSN betweenness was also a significant contributor to forecasting the trade value growth. The results were qualitatively the same when we replaced the countrywise trade value by the import value or export value. Furthermore, we found that adding either GLSN betweenness or GLSN connectivity to the gravity model improved its ability in estimating the bilateral trade value between them. These results support a long-standing view in maritime economics, which has yet to be directly tested, that countries that are more strongly integrated into the global maritime transportation network have better access to global markets and thus greater trade opportunities [9]. A previous study has supported that improving bilateral connectivity in liner shipping can facilitate the bilateral trade between two countries, suggesting the influence of liner shipping connectivity on international

trade at the country pair level [53]. The present study provides insights into understanding such influences at the country level, by focusing on the liner shipping connectivity of individual countries.

The GLSN connectivity and GLSN betweenness are variants of node's degree centrality and betweenness centrality, respectively. We used the information on the nationality of ports and service routes (i.e. the list of ports included in each service route) to inform the two indices. The GLSN betweenness supports the structural hole theory dictating in the present context that ports possessing more structural holes in the GLSN would provide the country with greater trading opportunities in global markets. A structural hole is the absence of a tie among a pair of nodes in the ego-centric network [54]. An established proposition in social network analysis is that nodes with many structural holes are strong performers in competitive settings [55], taking advantage of the missing connections between its neighbouring nodes. We defined the GLSN betweenness, in particular with $L_{max} = 2$, by counting the so-called valid shortest paths of length 2, which are equivalent to open triads composed of three ports all of which are located in different countries. Because the valid shortest path with $L_{max} = 2$ requires that a port of the focal country is located between two foreign ports of different countries that are not adjacent to each other, the GLSN betweenness quantifies the number of structural holes that the given country's ports have. The strong correspondence between the GLSN betweenness and the country's trade value suggests that occupying structural holes between foreign ports may be advantageous in international trade. The GLSN betweenness may reflect the extent to which a country's ports serve as trans-shipment centres for cargo transportation between ports of different countries.

There are various maritime transport modes serving cargo transportation, i.e. bulk cargo shipping, general cargo shipping and liner shipping [56]. Among them, only liner shipping involves inter-port trans-shipment activities (i.e. ports mediate shipping between other ports), as specifically designed by liner shipping companies. For the other maritime transport modes, it is common that cargos are directly shipped from a port of origin to a port of destination, such that a service route normally consists of two ports. Therefore, the equivalents of the GLSN connectivity and GLSN betweenness for maritime transport networks of these different modes are not expected to be strong indicators of international trade values. In fact, liner shipping accounts for more than 70% of the cargo value transported by sea [4]. Therefore, we consider that our finding provides promising tools to interest groups, such as shipping carriers, international trading companies, economic think tanks, national governments, and international organizations such as the UNCTAD and the World Bank, for measuring and predicting international trade status of countries.

Establishing a causal relationship between GLSN metrics and international trade requires longitudinal analyses of maritime and economic data. Revealing such a causal relationship is expected to have a large socioeconomic impact because GLSN data is usually released much earlier than trade data. In fact, shipping companies pre-release their liner shipping service routes even one year prior to making voyages. In-depth analyses of causality between GLSNs and international trade are left as future work. Another limitation of this study is that we have not considered the directionality of the edges in the GLSN. The information on the sequence of port calls in each service route is available in our dataset. However, it remains challenging to infer the directionality of each inter-port connection in a service route, and hence the direction of edges between ports and between countries. This is because there may exist cargo transportation between two ports that are not sequentially called at. For example, in a circular route that calls at ports A, B, C, D, E, C and A in this order, any of the five ports may send cargos to any other port, rendering the estimation of direct edges difficult. Directed GLSNs, if one can reasonably estimate them, may contribute to improving the accuracy of describing import and export trade values.

In conclusion, we have provided evidence that countries' positions in the GLSN are associated with their international trade status. Our results are expected to seed further research towards quantitatively understanding of the interplay between the structure of maritime shipping networks and international trade. For example, the global liner shipping system is evolving over time. Therefore, investigation of the dynamics of the GLSN and how they are related to dynamic changes in countries' international trade status warrants future work. We also remark that liner shipping, though dominant in terms of the cargo value, is not the only mode of maritime shipping that facilitates the seaborne trade worldwide. Maritime shipping networks that incorporate multiple major shipping modes (e.g. liner shipping, dry bulk shipping and oil shipping) [57,58] await further exploration.

Data accessibility. Data and relevant code for this research work are stored in GitHub: https://github.com/Network-Maritime-Complexity/GLSN-and-international-trade; and have been archived within the Zenodo repository:

https://doi.org/10.5281/zenodo.4018584. The raw data on world liner shipping service routes were provided by Alphaliner (https://www.alphaliner.com/), which is a world's leading provider of data in liner shipping, and were used under the licence for the current study. Therefore, these raw data are not publicly available.

Authors' contributions. M.X. and N.M. designed research and wrote the paper; M.X. performed research; Q.P. generated code; M.X., Q.P., H.X. and N.M. analysed data. All authors gave final approval for publication.

Competing interests. The authors declare no competing interests.

Funding. M.X. was supported by the Fundamental Research Funds for the Central Universities (China), and the China Postdoctoral Science Foundation (grant no. 2017M621141). H.X. was supported by National Natural Science Foundation of China (grant nos. 71871042, 71533001 and 71421001), and the Scientific and Technological Innovation Foundation of Dalian (China) (grant no. 2018J11CY009).

Acknowledgements. M.X. thanks Teruyoshi Kobayashi and Yi Niu for discussion. M.X. acknowledges the Alphaliner database for providing the data, the support provided by the Fundamental Research Funds for the Central Universities (China), and the support provided through China Postdoctoral Science Foundation. H.X. acknowledges the support provided through National Natural Science Foundation of China, and the Scientific and Technological Innovation Foundation of Dalian (China). N.M. acknowledges the financial support from Dalian University of Technology regarding visits for scientific collaboration.

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
