## [Reviewer comments · Royal Society Open Science]

Review History

RSOS-200386.R0 (Original submission)

Review form: Reviewer 1

Is the manuscript scientifically sound in its present form?

Yes

Are the interpretations and conclusions justified by the results?

Yes

Is the language acceptable?

Yes

Do you have any ethical concerns with this paper?

No

Have you any concerns about statistical analyses in this paper?

No

Recommendation?

Major revision is needed (please make suggestions in comments)

Comments to the Author(s)

Dear authors,

I think the paper is basically sound and none of the conclusions drawn are surprising. However, in order to add deeper knowledge to the field, can you please clarify the following:

1. [Data] Does your data include only port of origin and destination or also refuelling and midway load/unloading ports (I think from Fig 3, I can see it does, but I am not sure). For example, I understand that (as an example) Singapore is a very high betweenness port in the sense that few ships start or end there, but many pass through. This is similar for underground/subways (few stations are start/final stops), but different to airports (where very few are multi-hops). This would dramatically change your conclusion I imagine. I note this particularly, because your visualisation on Fig 1. doesn't show coastal hugging or many multi-hops it seems?
2. [Data] How complete is your data? I ask from 2 perspectives. First is the ship class, what classes are covered and how complete is this (cargo size, cargo type)? Second is the temporal and frequency representation (how many times per year)?
3. [Method] Betweenness assumes that the existing network cannot be rewired and agents are somehow restricted to exploit the shortest paths along the current network. However, we know that ships can sail from A to B, even if no previous connections exist. How do you explain for this?
4. [Knowledge] What knowledge is gained? Your argument is that the position of a port/country in the GLSN is causally contributing to its international trade value. But how do we know it is not vice versa. There are cases where your argument is obvious: Singapore / Panama. There are cases where there is historical precedence rather than geographical precedence (strong naval traditions of UK and Netherlands). My point is: (1) we have the quantitative ground truth which is the trade data, (2) we have qualitative truth through simple geographical and historical arguments on strengths of different ports --- as such, what is the added value of GLSN? Is it to quantify this in a lower dimensional way?

Review form: Reviewer 2

Is the manuscript scientifically sound in its present form?

No

Are the interpretations and conclusions justified by the results?

No

Is the language acceptable?

Yes

Do you have any ethical concerns with this paper?

No

Have you any concerns about statistical analyses in this paper?

Yes

Recommendation?

Major revision is needed (please make suggestions in comments)

Comments to the Author(s)

Review 'Estimating international trade status of countries from global liner shipping networks'

I would like to congratulate the authors for coming up with an interesting and timely research idea, making use of global shipping data and advances in network analysis to analyse the role of maritime network characteristics in driving global trade. The authors show a clear competence in working with networks and have come up with an interesting new indicator of connectivity, which they show, is better than the previously derived indicators (e.g. UNCTAD's LSCI). The authors go on in setting up a set of regression models to explain trade, GDP, changes in trade and bilateral trade. This is my main objection with the approach taken, as the purpose of a regression analysis is not just to "find the best fitting model", but setting up a hypothesis to prove. In my opinion, whether or not the relationships are causal or not is hard to judge, because the indicator probably correlates well with other indicators that explain trade (for instance GDP, as included in the gravity equation, and shown to be highly correlated with the GLSN indicator). Therefore, it is hard to answer the question: Does an increase in the GLSN indicator cause an increase in trade, or does higher GDP cause an increased GLSN and hence higher trade value? The regression analysis for GDP and trade changes are not relevant in my opinion, because of the abovementioned points. For the comparison with the gravity equation, I would highly recommend the authors to change their framing from: 'our indicators performs better than the gravity equation' to 'how does including our parameter in the gravity equation improve bilateral trade estimates'. The latter has much wider policy implications, if proved, since it means that the current friction formulation in the gravity equation does not capture the variability in trade, which the GLSN indicator might improve (for instance, are countries directly connected, or is transshipment necessary?). Therefore, I would re-do some of the analysis in line with previous work (Fugazza and Hoffman et al. 2017) and evaluate whether or not your indicator is better in improving the gravity equation than their indicator.

Major corrections:

-Please have a native speaker check the manuscript for English grammar and structure of sentences.

-P4.L32: The aim of the present study is mentioned, but there is not much information why this hypothesis is proposed and what evidence already exists to support this? This should be introduced much more clearly here. Also I would consider another term for 'middleman'

-P5.L10 The use of an undirected network may be flawed, in particular when looking at liner shipping where a lot of liner schedules have a pendulum network designs. I understand the choice of using an undirected network, but do at least reflect on this in this section or the discussion.

-It seems like the author have missed a major research article related to their work: Fugazza, M. and Hoffmann, J. (2017) 'Liner shipping connectivity as determinant of trade', *Journal of Shipping and Trade*, 2(1), pp. 1-18. doi: 10.1186/s41072-017-0019-5.

Please make sure to include it and reflect on the differences in method and results.

-I am not convinced by the relationship found in 2.5. I don't see a clear explanation for the fact why the Gb2015 would be essential to explain growth, apart from the fact that trade is a global phenomenon and that a lot of countries see a similar increase in trade, hence the importance of Tv2015. I think the authors should think a bit deeper about the implications of setting up this model. It would be interesting to see if changes in the Gb2015 are an indicator of changes in trade, which could mean that if ports are becoming increasingly connected to other ports, it would

explain the a larger increment in trade than a country where no changes in the Gb2015 were observed.

-Section 2.6. Instead of comparing the gravity model and GLSN, it would be more interesting to see what happens if the GLSN is added to the gravity equation: e.g. can the trade between countries be better estimated by an improved maritime connectivity value?. I think this has much broader policy implications, as it means that the standard gravity equation, that relies on distance between countries as friction, is overly simplistic if these countries are not well connected (e.g. do not have a direct liner connection.).

-The link to the GitHub does not yet exist. Please do create this before submitting for review.

Minor corrections

-P3.L29: "important to" not good English. Change this sentence.

-P3.L29: with maritime countries, you mean countries that are not landlocked?

-P3.L37: "as various proposed indices quantify". This is unclear, either change or remove.

-P3.L50: Sentence starting with "The LSCI" is very long, please break up.

-P4.L13: 'even landlocked' is not good English.

-P4.L29: "and thus global economy", should be "trade, and thus the global economy, but do". Also this sentence is not entirely clear as it was just mentioned how the LSBCI is the connectivity between countries. I think what you are trying to say is the interconnectivity of all ports and countries?

-P5.L11: Why six different edge weighting methods? What are the weights of this network, the frequency, or the actual cargo trade?

From the method section this seems to be TEU. At least mention this briefly in the main text.

-P6.L53: I am not sure you need to justify keeping the LSCI in.

-Results section: Would it be possible to make some maps with a few of the indicators and put them in the Supplement? This would help to give an idea of the differences and spatial distribution between the weight methods.

-L7.54: Please change "among the survivors" to "Among the remaining model configurations" or something similar.

-P8.L12: The LSCI is excluded because of high collinearity? Or because it does not predict the results well enough?

-P8.L20: Great that you decided to do the same analysis for different trade metrics. Maybe it is good to reflect a bit on this; why the net export is less well correlated/explained by the model (which makes a lot of sense, but good to mention).

-P9.L46: You don't have to give a definition of how GDP is calculated.

-P9.L39: I am wondering whether it is relevant to explain GDP using the network indicators, as GDP is most likely indirectly correlated to the indicators through the trade value (as also included in the gravity equation). Or there is any other way in which the indicators are related to GDP, but I don't see what this is. This also has a lot to do with the question whether or not

growth in GDP leads to more investment in port infrastructure and better connectivity, or if investment in port infrastructure lead to GDP growth. I would be careful making any claim about the relationship with GDP.

-P10.L29: Please rephrase the first sentence, I think you want to say “power on the changes in trade value over the years”.

-P13.L24 This last section is weak. I would like to see a concluding remark here, not a reference to other, non-relevant, literature. I think there is more relevant future work here related to the scope of the study.

Review form: Reviewer 3

Is the manuscript scientifically sound in its present form?

Yes

Are the interpretations and conclusions justified by the results?

No

Is the language acceptable?

Yes

Do you have any ethical concerns with this paper?

No

Have you any concerns about statistical analyses in this paper?

No

Recommendation?

Major revision is needed (please make suggestions in comments)

Comments to the Author(s)

Dear authors,

Thank you for submitting this manuscript. I think the work is interesting, but I had to read it twice before I started to understand all the things described in this piece. Also, think about the message you want to convey with this paper. There are many results now on the multivariate regressions, yet the introduction seems to incline that it is going to be a network analysis. Hence, I am a bit lost what the main message is of the paper when reading the results. Please find below some comments in which I provide some feedback on the manuscript.

- In the introduction, you say: “The overarching goal of the present study is to apply network analysis on shipping networks to derive useful quantitative knowledge about international trade and its growth for individual countries”. Is that true? The results primarily focus on the results of the (multivariate) regression analysis, which is, to my knowledge, not a network analysis. Don’t you just extract some network characteristics from the GLSN and use those in the regression analysis?

- In the introduction, you say: “The LSBCI was found to have a significant impact on South Africa’s bilateral trade flows with its trading partners”. Maybe I don’t understand the index, but how can an index have an impact on trade flows?

- Coming back to the point of understandability, I think the manuscript is not really written in a way that a non-expert can easily get the gist of the article. For instance, I would encourage to move the methods from the end to before the results. I think RSOS does not have that structure, which may have been an artifact from a previous submission to a different journal?

Without reading the methods first, it is very hard to understand the results. I would also move Figure 2 to the methods, it is not a result.

- Line 23-34 can be moved to the methods section
- What is the exact added value of the gravity model? The gravity model is only briefly explained in the methods section, and no words about the results of the gravity model in the discussion?
- I am not sure if RSOS requires a concluding section, but if so, please add one.

Overall, I think some of the main results are interesting and novel. I see the merit of this paper, but the way it's written requires more work. It's now primarily only easy to read for scholars in the maritime economics research field. As this is an interdisciplinary journal, I strongly suggest to rewrite it in such a way that the results are interesting and understandable to a more wider range of people.

Decision letter (RSOS-200386.R0)

Dear Dr Xu,

The editors assigned to your paper ("Estimating international trade status of countries from global liner shipping networks") have now received comments from reviewers. We would like you to revise your paper in accordance with the referee and Associate Editor suggestions which can be found below (not including confidential reports to the Editor). Please note this decision does not guarantee eventual acceptance.

Please submit a copy of your revised paper before 05-Aug-2020. Please note that the revision deadline will expire at 00.00am on this date. If we do not hear from you within this time then it will be assumed that the paper has been withdrawn. In exceptional circumstances, extensions may be possible if agreed with the Editorial Office in advance. We do not allow multiple rounds of revision so we urge you to make every effort to fully address all of the comments at this stage. If deemed necessary by the Editors, your manuscript will be sent back to one or more of the original reviewers for assessment. If the original reviewers are not available, we may invite new reviewers.

- Data accessibility

<http://datadryad.org/submit?journalID=RSOS&manu=RSOS-200386>

- Competing interests

- Authors' contributions

- Acknowledgements

- Funding statement

Kind regards,
Andrew Dunn
Royal Society Open Science Editorial Office

on behalf of Dr Cecilia Mascolo (Associate Editor) and Marta Kwiatkowska (Subject Editor)
 openscience@royalsociety.org

Associate Editor's comments (Dr Cecilia Mascolo):

Associate Editor: 1

Comments to the Author:

The reviewers unanimously suggest a major revision based on presentation and methodological issues. We hope the comments are useful in improving the manuscript.

Comments to Author:

Reviewers' Comments to Author:

Reviewer: 1

Comments to the Author(s)

Dear authors,

I think the paper is basically sound and none of the conclusions drawn are surprising. However, in order to add deeper knowledge to the field, can you please clarify the following:

1. [Data] Does your data include only port of origin and destination or also refuelling and midway load/unloading ports (I think from Fig 3, I can see it does, but I am not sure). For example, I understand that (as an example) Singapore is a very high betweenness port in the sense that few ships start or end there, but many pass through. This is similar for underground/subways (few stations are start/final stops), but different to airports (where very few are multi-hops). This would dramatically change your conclusion I imagine. I note this particularly, because your visualisation on Fig 1. doesn't show coastal hugging or many multi-hops it seems?
2. [Data] How complete is your data? I ask from 2 perspectives. First is the ship class, what classes are covered and how complete is this (cargo size, cargo type)? Second is the temporal and frequency representation (how many times per year)?
3. [Method] Betweenness assumes that the existing network cannot be rewired and agents are somehow restricted to exploit the shortest paths along the current network. However, we know that ships can sail from A to B, even if no previous connections exist. How do you explain for this?
4. [Knowledge] What knowledge is gained? Your argument is that the position of a port/country in the GLSN is causally contributing to its international trade value. But how do we know it is not vice versa. There are cases where your argument is obvious: Singapore / Panama. There are cases where there is historical precedence rather than geographical precedence (strong naval traditions of UK and Netherlands). My point is: (1) we have the quantitative ground truth which is the trade data, (2) we have qualitative truth through simple geographical and historical arguments on strengths of different ports --- as such, what is the added value of GLSN? Is it to quantify this in a lower dimensional way?

Reviewer: 2

Comments to the Author(s)

Review 'Estimating international trade status of countries from global liner shipping networks'

I would like to congratulate the authors for coming up with an interesting and timely research idea, making use of global shipping data and advances in network analysis to analyse the role of maritime network characteristics in driving global trade. The authors show a clear competence in working with networks and have come up with an interesting new indicator of connectivity, which they show, is better than the previously derived indicators (e.g. UNCTAD's LSCI). The authors go on in setting up a set of regression models to explain trade, GDP, changes in trade and bilateral trade. This is my main objection with the approach taken, as the purpose of a regression analysis is not just to "find the best fitting model", but setting up a hypothesis to prove. In my opinion, whether or not the relationships are causal or not is hard to judge, because the indicator probably correlates well with other indicators that explain trade (for instance GDP, as included in the gravity equation, and shown to be highly correlated with the GLSN indicator). Therefore, it is hard to answer the question: Does an increase in the GLSN indicator cause an increase in trade, or does higher GDP cause an increased GLSN and hence higher trade value? The regression analysis for GDP and trade changes are not relevant in my opinion, because of the abovementioned points. For the comparison with the gravity equation, I would highly recommend the authors to change their framing from: 'our indicators performs better than the gravity equation' to 'how does including our parameter in the gravity equation improve bilateral trade estimates'. The latter has much wider policy implications, if proved, since it means that the current friction formulation in the gravity equation does not capture the variability in trade, which the GLSN indicator might improve (for instance, are countries directly connected, or is transshipment necessary?). Therefore, I would re-do some of the analysis in line with previous work (Fugazza and Hoffman et al. 2017) and evaluate whether or not your indicator is better in improving the gravity equation than their indicator.

Major corrections:

-Please have a native speaker check the manuscript for English grammar and structure of sentences.

-P4.L32: The aim of the present study is mentioned, but there is not much information why this hypothesis is proposed and what evidence already exists to support this? This should be introduced much more clearly here. Also I would consider another term for 'middleman'

-P5.L10 The use of an undirected network may be flawed, in particular when looking at liner shipping where a lot of liner schedules have a pendulum network designs. I understand the choice of using an undirected network, but do at least reflect on this in this section or the discussion.

-It seems like the author have missed a major research article related to their work: Fugazza, M. and Hoffmann, J. (2017) 'Liner shipping connectivity as determinant of trade', *Journal of Shipping and Trade*. *Journal of Shipping and Trade*, 2(1), pp. 1-18. doi: 10.1186/s41072-017-0019-5.

Please make sure to include it and reflect on the differences in method and results.

-I am not convinced by the relationship found in 2.5. I don't see a clear explanation for the fact why the Gb2015 would be essential to explain growth, apart from the fact that trade is a global phenomenon and that a lot of countries see a similar increase in trade, hence the importance of Tv2015. I think the authors should think a bit deeper about the implications of setting up this model. It would be interesting to see if changes in the Gb2015 are an indicator of changes in trade, which could mean that if ports are becoming increasingly connected to other ports, it would explain the a larger increment in trade than a country where no changes in the Gb2015 were observed.

-Section 2.6. Instead of comparing the gravity model and GLSN, it would be more interesting to see what happens if the GLSN is added to the gravity equation: e.g. can the trade between

countries be better estimated by an improved maritime connectivity value? I think this has much broader policy implications, as it means that the standard gravity equation, that relies on distance between countries as friction, is overly simplistic if these countries are not well connected (e.g. do not have a direct liner connection.).

-The link to the GitHub does not yet exist. Please do create this before submitting for review.

Minor corrections

-P3.L29: "important to" not good English. Change this sentence.

-P3.L29: with maritime countries, you mean countries that are not landlocked?

-P3.L37: "as various proposed indices quantify". This is unclear, either change or remove.

-P3.L50: Sentence starting with "The LSCI" is very long, please break up.

-P4.L13: 'even landlocked' is not good English.

-P4.L29: "and thus global economy", should be "trade, and thus the global economy, but do". Also this sentence is not entirely clear as it was just mentioned how the LSBCI is the connectivity between countries. I think what you are trying to say is the interconnectivity of all ports and countries?

-P5.L11: Why six different edge weighting methods? What are the weights of this network, the frequency, or the actual cargo trade?

From the method section this seems to be TEU. At least mention this briefly in the main text.

-P6.L53: I am not sure you need to justify keeping the LSCI in.

-Results section: Would it be possible to make some maps with a few of the indicators and put them in the Supplement? This would help to give an idea of the differences and spatial distribution between the weight methods.

-L7.54: Please change "among the survivors" to "Among the remaining model configurations" or something similar.

-P8.L12: The LSCI is excluded because of high collinearity? Or because it does not predict the results well enough?

-P8.L20: Great that you decided to do the same analysis for different trade metrics. Maybe it is good to reflect a bit on this; why the net export is less well correlated/explained by the model (which makes a lot of sense, but good to mention).

-P9.L46: You don't have to give a definition of how GDP is calculated.

-P9.L39: I am wondering whether it is relevant to explain GDP using the network indicators, as GDP is most likely indirectly correlated to the indicators through the trade value (as also included in the gravity equation). Or there is any other way in which the indicators are related to GDP, but I don't see what this is. This also has a lot to do with the question whether or not growth in GDP leads to more investment in port infrastructure and better connectivity, or if investment in port infrastructure lead to GDP growth. I would be careful making any claim about the relationship with GDP.

-P10.L29: Please rephrase the first sentence, I think you want to say "power on the changes in trade value over the years".

-P13.L24 This last section is weak. I would like to see a concluding remark here, not a reference to other, non-relevant, literature. I think there is more relevant future work here related to the scope of the study.

Reviewer: 3

Comments to the Author(s)

Dear authors,

Thank you for submitting this manuscript. I think the work is interesting, but I had to read it twice before I started to understand all the things described in this piece. Also, think about the message you want to convey with this paper. There are many results now on the multivariate regressions, yet the introduction seems to incline that it is going to be a network analysis. Hence, I am a bit lost what the main message is of the paper when reading the results. Please find below some comments in which I provide some feedback on the manuscript.

- In the introduction, you say: "The overarching goal of the present study is to apply network analysis on shipping networks to derive useful quantitative knowledge about international trade and its growth for individual countries". Is that true? The results primarily focus on the results of the (multivariate) regression analysis, which is, to my knowledge, not a network analysis. Don't you just extract some network characteristics from the GLSN and use those in the regression analysis?

- In the introduction, you say: "The LSBCI was found to have a significant impact on South Africa's bilateral trade flows with its trading partners". Maybe I don't understand the index, but how can an index have an impact on trade flows?

- Coming back to the point of understandability, I think the manuscript is not really written in a way that a non-expert can easily get the gist of the article. For instance, I would encourage to move the methods from the end to before the results. I think RSOS does not have that structure, which may have been an artifact from a previous submission to a different journal? Without reading the methods first, it is very hard to understand the results. I would also move Figure 2 to the methods, it is not a result.

- Line 23-34 can be moved to the methods section

- What is the exact added value of the gravity model? The gravity model is only briefly explained in the methods section, and no words about the results of the gravity model in the discussion?

- I am not sure if RSOS requires a concluding section, but if so, please add one.

Overall, I think some of the main results are interesting and novel. I see the merit of this paper, but the way it's written requires more work. It's now primarily only easy to read for scholars in the maritime economics research field. As this is an interdisciplinary journal, I strongly suggest to rewrite it in such a way that the results are interesting and understandable to a more wider range of people.

Author's Response to Decision Letter for (RSOS-200386.R0)

See Appendix A.

RSOS-200386.R1 (Revision)

Review form: Reviewer 1

Is the manuscript scientifically sound in its present form?

Yes

Are the interpretations and conclusions justified by the results?

Yes

Is the language acceptable?

Yes

Do you have any ethical concerns with this paper?

No

Have you any concerns about statistical analyses in this paper?

No

Recommendation?

Accept as is

Comments to the Author(s)

The authors addressed all my comments.

Review form: Reviewer 2

Is the manuscript scientifically sound in its present form?

Yes

Are the interpretations and conclusions justified by the results?

Yes

Is the language acceptable?

Yes

Do you have any ethical concerns with this paper?

No

Have you any concerns about statistical analyses in this paper?

No

Recommendation?

Accept with minor revision (please list in comments)

Comments to the Author(s)

Review 'Estimating international trade status of countries from global liner shipping networks'

I am very satisfied with the way the authors have responded to my comments and the adjustments that are made. I still have a few style/grammar points to improve the manuscript. Additionally, I would like to emphasize that the main message that you re-iterate in the discussion: "These results support a long-standing view in maritime economics, which has yet

been directly tested, that countries that are more strongly integrated into the global maritime transportation network have better access to global markets and thus greater trade opportunities" is really the main overall message and hypothesis here in my opinion. I think this is more important and clear (in terms of communication) compared to the main hypothesis in the article proposed now: "We hypothesize that the role of a port or country as broker to mediate liner shipping between different countries is correlated with the importance of the port or country in international trade." You can consider rewriting the article a little bit so that the first sentence is the main message, and the broker role is the means by which integration is facilitated in practise. I think in this way it could improve the communication of the article and its readability for a broader audience.

P2L45: "Vessels travel from a port of one country to another on networks of ports to carry cargos, which to countries' international trade values." Please rewrite this sentence as it is unclear. Something in the order of: "Cargo loaded vessels travel from one country's port to another via an underlying port-to-port transport network."

P2L52: "international trade value" should be "bilateral trade flows."

P2L82: "bilateral trade values" and "trading partners" is double, please remove "with its trading partners" or write: "found to be correlated with SA's trade with its trading partners."

P3L98: I don't understand the word "orthogonal" in this sentence. Please change.

P3L116: Please rephrase this sentence. First, I don't understand what is meant with centrality measures that neglect the nationality of the ports or service routes. Second, why are these centrality measures poor indicators? I think I get what you are saying but this is quite a critical point for doing this research so I would like it emphasized more.

P3L121: "quantify roles which each country plays as broker", change to: "quantify each country's role as broker in international maritime transport."

P4L126: "better that of the LSCI." I would phrase a bit more nuanced, such as: "is found to perform better than previously established liner connectivity metrics such as the LSCI."

P4L131: "our previous studies", to, "previous work."

P4L136: "The types of ship that data set covers" to 'the vessel types included in the dataset are'

P14L455: "These two variables, but not any other, were also"

P15L515: Although it is great that you ran all combinations, a reader is only interested in the improvements compared to the original gravity model. Could you change the table to the original gravity model as baseline, and then followed by the extended models only. This makes it much easier to quickly see the improvements of it.

P16L518: replace 'stems' with 'originates' or 'is shaped by multiple'.

P16:519: replace 'seek for profits', 'which primarily aim to maximise profits in a decentralized manner'.

P16L548: I would like to have this explanation of the structural hole theory in the introduction instead of discussion, because the structural hole theory was introduced there but not explained. It seems to be relevant for how you have constructed the centrality indicators, which is therefore essential information.

Decision letter (RSOS-200386.R1)

Dear Dr Xu

On behalf of the Editors, we are pleased to inform you that your Manuscript RSOS-200386.R1 "Estimating international trade status of countries from global liner shipping networks" has been accepted for publication in Royal Society Open Science subject to minor revision in accordance with the referees' reports. Please find the referees' comments along with any feedback from the Editors below my signature.

Please submit your revised manuscript and required files (see below) no later than 7 days from today's (ie 03-Sep-2020) date. Note: the ScholarOne system will 'lock' if submission of the revision is attempted 7 or more days after the deadline. If you do not think you will be able to meet this deadline please contact the editorial office immediately.

on behalf of Prof Marta Kwiatkowska (Subject Editor)
openscience@royalsociety.org

Reviewer comments to Author:
Reviewer: 2

Comments to the Author(s)
Review 'Estimating international trade status of countries from global liner shipping networks'

I am very satisfied with the way the authors have responded to my comments and the adjustments that are made. I still have a few style/grammar points to improve the manuscript. Additionally, I would like to emphasize that the main message that you re-iterate in the discussion: "These results support a long-standing view in maritime economics, which has yet been directly tested, that countries that are more strongly integrated into the global maritime

transportation network have better access to global markets and thus greater trade opportunities” is really the main overall message and hypothesis here in my opinion. I think this is more important and clear (in terms of communication) compared to the main hypothesis in the article proposed now: “We hypothesize that the role of a port or country as broker to mediate liner shipping between different countries is correlated with the importance of the port or country in international trade.” You can consider rewriting the article a little bit so that the first sentence is the main message, and the broker role is the means by which integration is facilitated in practise. I think in this way it could improve the communication of the article and its readability for a broader audience.

P2L45: “Vessels travel from a port of one country to another on networks of ports to carry cargos, which to countries’ international trade values.” Please rewrite this sentence as it is unclear. Something in the order of: “Cargo loaded vessels travel from one country’s port to another via an underlying port-to-port transport network.”

P2L52: “international trade value” should be “bilateral trade flows.”

P2L82: “bilateral trade values” and “trading partners” is double, please remove “with its trading partners” or write: “found to be correlated with SA’s trade with its trading partners.”

P3L98: I don’t understand the word “orthogonal” in this sentence. Please change.

P3L116: Please rephrase this sentence. First, I don’t understand what is meant with centrality measures that neglect the nationality of the ports or service routes. Second, why are these centrality measures poor indicators? I think I get what you are saying but this is quite a critical point for doing this research so I would like it emphasized more.

P3L121: “quantify roles which each country plays as broker”, change to: “quantify each country’s role as broker in international maritime transport.”

P4L126: “better that of the LSCI.” I would phrase a bit more nuanced, such as: “is found to perform better than previously established liner connectivity metrics such as the LSCI.”

P4L131: “our previous studies”, to, “previous work.”

P4L136: “The types of ship that data set covers” to ‘the vessel types included in the dataset are’

P14L455: “These two variables, but not any other, were also”

P15L515: Although it is great that you ran all combinations, a reader is only interested in the improvements compared to the original gravity model. Could you change the table to the original gravity model as baseline, and then followed by the extended models only. This makes it much easier to quickly see the improvements of it.

P16L518: replace ‘stems’ with ‘originates’ or ‘ is shaped by multiple’.

P16:519: replace ‘seek for profits’, ‘which primarily aim to maximise profits in a decentralized manner’.

P16L548: I would like to have this explanation of the structural hole theory in the introduction instead of discussion, because the structural hole theory was introduced there but not explained. It seems to be relevant for how you have constructed the centrality indicators, which is therefore essential information.

Reviewer: 1

Comments to the Author(s)

The authors addressed all my comments.

===PREPARING YOUR MANUSCRIPT===

===PREPARING YOUR REVISION IN SCHOLARONE===

- 1) One version identifying all the changes that have been made (for instance, in coloured highlight, in bold text, or tracked changes);
 - 2) A 'clean' version of the new manuscript that incorporates the changes made, but does not highlight them.
 - An individual file of each figure (EPS or print-quality PDF preferred [either format should be produced directly from original creation package], or original software format).
 - An editable file of each table (.doc, .docx, .xls, .xlsx, or .csv).
 - An editable file of all figure and table captions.
- Note: you may upload the figure, table, and caption files in a single Zip folder.
- Any electronic supplementary material (ESM).
 - If you are requesting a discretionary waiver for the article processing charge, the waiver form must be included at this step.
 - If you are providing image files for potential cover images, please upload these at this step, and inform the editorial office you have done so. You must hold the copyright to any image provided.
 - A copy of your point-by-point response to referees and Editors. This will expedite the preparation of your proof.

- Ensure that your data access statement meets the requirements at <https://royalsociety.org/journals/authors/author-guidelines/#data>. You should ensure that you cite the dataset in your reference list. If you have deposited data etc in the Dryad repository, please only include the 'For publication' link at this stage. You should remove the 'For review' link.
- If you are requesting an article processing charge waiver, you must select the relevant waiver option (if requesting a discretionary waiver, the form should have been uploaded at Step 3 'File upload' above).
- If you have uploaded ESM files, please ensure you follow the guidance at <https://royalsociety.org/journals/authors/author-guidelines/#supplementary-material> to include a suitable title and informative caption. An example of appropriate titling and captioning may be found at https://figshare.com/articles/Table_S2_from_Is_there_a_trade-off_between_peak_performance_and_performance_breadth_across_temperatures_for_aerobic_scope_in_teleost_fishes_/3843624.

Author's Response to Decision Letter for (RSOS-200386.R1)

See Appendix B.

Decision letter (RSOS-200386.R2)

Dear Dr Xu,

It is a pleasure to accept your manuscript entitled "Estimating international trade status of countries from global liner shipping networks" in its current form for publication in Royal Society Open Science.

on behalf of Prof Marta Kwiatkowska (Subject Editor)
openscience@royalsociety.org

Associate Editor Comments to Author:

Thank you kindly for submitting your revised manuscript. I believe you have sufficiently responded to all remaining referee comments, and that your revised manuscript should be accepted as is. Thank you for choosing Royal Society Open Science for your study.

Appendix A

Associate Editor's comments (Dr Cecilia Mascolo):

Associate Editor: 1

Comments to the Author:

The reviewers unanimously suggest a major revision based on presentation and methodological issues. We hope the comments are useful in improving the manuscript.

We thank the editor and the reviewers for evaluating our manuscript. We have improved the manuscript in accordance with the reviewers' valuable comments. To assist the editor and the reviewers in checking our revision, we provide the revised manuscript in which the modified and the added text is in blue.

Reviewer 1

Dear authors,

I think the paper is basically sound and none of the conclusions drawn are surprising. However, in order to add deeper knowledge to the field, can you please clarify the following:

We are glad to hear an overall positive assessment. We amended the manuscript in accordance with your valuable comments as follows.

1. [Data] Does your data include only port of origin and destination or also refuelling and midway load/unloading ports (I think from Fig 3, I can see it does, but I am not sure). For example, I understand that (as an example) Singapore is a very high betweenness port in the sense that few ships start or end there, but many pass through. This is similar for underground/subways (few stations are start/final stops), but different to airports (where very few are multi-hops). This would dramatically change your conclusion I imagine. I note this particularly, because your visualisation on Fig 1. doesn't show coastal hugging or many multi-hops it seems?

The data on each service route include ports of origin and destination, midway load/unloading ports, and as well as the order in which the ports are visited. But the data do not include refueling ports. To clarify, we added text to explain these facts (lines 142-144, page 4).

2. [Data] How complete is your data? I ask from 2 perspectives. First is the ship class, what classes are covered and how complete is this (cargo size, cargo type)? Second is the temporal and frequency representation (how many times per year)?

Regarding the first question, we have added the following text (lines 136-142, page 4). As we understand it, the reviewer's term "ship class" refers to "ship type". Please note that we do not have further information about the data.

"The types of ship that data set covers are full-container vessels and multi-purpose vessels. We use the information on 1316 international liner shipping service routes (service route for short), all of

which were deployed with full-container vessels. We exclude service routes with multi-purpose vessels because service routes with full-container vessels are the most common in liner shipping practice [39]. Therefore, the cargo type is containerized cargo. The information on the cargo size for each ship is not available.”

Regarding the second question, the data was obtained at a given point of time, i.e., April, 2015. We have added this information to the main text (line 130, page 4). Therefore, we do not have temporal or frequency information. To discuss this issue as a future research direction, we have added the following text (lines 596-599, page 17) to the final paragraph of the manuscript.

“For example, the global liner shipping system is evolving over time. Therefore, investigation of the dynamics of the GLSN and how they are related to dynamic changes in countries’ international trade status warrants future work.”

3. [Method] Betweenness assumes that the existing network cannot be rewired and agents are somehow restricted to exploit the shortest paths along the current network. However, we know that ships can sail from A to B, even if no previous connections exist. How do you explain for this?

As the reviewer mentioned, the betweenness (no matter which betweenness measure used in the present manuscript is concerned) is a node centrality measure (in this manuscript) for static networks (i.e., no rewiring). If ships can sail from A to B, they are contained in the same route, therefore, there is an edge between A and B according to our definition of the edge. Therefore, it does not happen that ships sail from A to B when there is no edge between A and B.

4. [Knowledge] What knowledge is gained? Your argument is that the position of a port/country in the GLSN is causally contributing to its international trade value. But how do we know it is not vice versa. There are cases where your argument is obvious: Singapore / Panama. There are cases where there is historical precedence rather than geographical precedence (strong naval traditions of UK and Netherlands).

First, we would like to highlight the following knowledge gained from the present study: This study reveals that a country’s position in the GLSN can be a strong signature of the country’s international trade value, thus quantifying the correspondence between the proposed GLSN connectivity and international trade at the country level. In the previous version, we described this in the first paragraph of the Discussion section (lines 522–529, page 16, in the present version), which reads as follows:

“Based on a comprehensive port-level global liner shipping network data set, we constructed GLSNs and proposed network indicators to quantify individual countries’ positions in the GLSN. We showed that a country’s position in the GLSN was a strong signature of the country’s international trade value. In particular, we proposed the GLSN connectivity and GLSN betweenness indices,

which one can calculate from local information about the network around the ports of the focal country. The two indices explained the trade value fairly well. The GLSN betweenness was also a significant contributor to forecasting the trade value growth.”

Second, we agree with the reviewer that the knowledge gained in the present study should not be claimed as the causal influence of the structure of GLSNs on the country’s trade value. In fact, we never claimed so throughout the manuscript and did discuss this issue in the previous version in the second-to-last paragraph of the Discussion section (lines 577-582, page 17 in the present version); it reads:

“Establishing a causal relationship between GLSN metrics and international trade requires longitudinal analyses of maritime and economic data. Revealing such a causal relationship is expected to have a large socioeconomic impact because GLSN data is usually released much earlier than trade data. In fact, shipping companies pre-release their liner shipping service routes even one year prior to making voyages. In-depth analyses of causality between GLSNs and international trade are left as future work.”

My point is: (1) we have the quantitative ground truth which is the trade data, (2) we have qualitative truth through simple geographical and historical arguments on strengths of different ports --- as such, what is the added value of GLSN? Is it to quantify this in a lower dimensional way?

We agree with both (1) and (2). However, the knowledge produced by the present study is admittedly not qualitative (which may contribute to e.g. geographical and historical arguments on strengths of different ports such as Singapore and Panama) but quantitative. Specifically, the GLSN indicators proposed in our study perform even better than the previously derived indicators (e.g. UNCTAD’s LSCI) in estimating countries’ international trade values. We stated this in several places in the manuscript (both the text that already existed in the previous version and the text we have added in this revision) (lines 50-53, page 2; lines 119-126, pages 3-4; lines 348-358, page 11; lines 522-529, page 16 in the present version).

To discuss the qualitative side, we wrote it in the third paragraph of the Discussion section in the previous version of the manuscript (lines 572–576, page 17 in the present version), which reads: “we consider that our finding provides promising tools to interest groups, such as shipping carriers, international trading companies, economic think tanks, national governments, and international organizations such as the UNCTAD and the World Bank, for measuring and predicting international trade status of countries.”

Reviewer 2

I would like to congratulate the authors for coming up with an interesting and timely research idea, making use of global shipping data and advances in network analysis

to analyse the role of maritime network characteristics in driving global trade. The authors show a clear competence in working with networks and have come up with an interesting new indicator of connectivity, which they show, is better than the previously derived indicators (e.g. UNCTAD's LSCI).

We are glad to hear an overall positive assessment by the reviewer.

The authors go on in setting up a set of regression models to explain trade, GDP, changes in trade and bilateral trade. This is my main objection with the approach taken, as the purpose of a regression analysis is not just to "find the best fitting model", but setting up a hypothesis to prove. In my opinion, whether or not the relationships are causal or not is hard to judge, because the indicator probably correlates well with other indicators that explain trade (for instance GDP, as included in the gravity equation, and shown to be highly correlated with the GLSN indicator). Therefore, it is hard to answer the question: Does an increase in the GLSN indicator cause an increase in trade, or does higher GDP cause an increased GLSN and hence higher trade value? The regression analysis for GDP and trade changes are not relevant in my opinion, because of the abovementioned points.

First, we agree that our results cannot tell whether or not the GLSN indicators have causal influence on countries' international trade status. We had already mentioned it in the penultimate paragraph of the Discussion section in the previous version of the manuscript (lines 577-582 on page 17 in the current version); it reads:

"Establishing a causal relationship between GLSN metrics and international trade requires longitudinal analyses of maritime and economic data. Revealing such a causal relationship is expected to have a large socioeconomic impact because GLSN data is usually released much earlier than trade data. In fact, shipping companies pre-release their liner shipping service routes even one year prior to making voyages. In-depth analyses of causality between GLSNs and international trade are left as future work."

Second, regarding the regression analysis for GDP and trade changes over time, we have provided further explanation to address your specific comments below. Please find our detailed reply on page 13 of this response letter regarding the GDP analysis and on page 9 regarding the analysis of trade changes over time.

For the comparison with the gravity equation, I would highly recommend the authors to change their framing from: 'our indicators performs better than the gravity equation' to 'how does including our parameter in the gravity equation improve bilateral trade estimates'. The latter has much wider policy implications, if proved, since it means that the current friction formulation in the gravity equation does not capture the variability in trade, which the GLSN indicator might improve (for instance, are countries directly connected, or is transshipment necessary?).

Therefore, I would re-do some of the analysis in line with previous work (Fugazza and Hoffman et al. 2017) and evaluate whether or not your indicator is better in improving the gravity equation than their indicator.

Thank you for insightful comments. We carried out additional analysis of the gravity model, including the version with UNCTAD's LSBCI (liner shipping bilateral connectivity index), which was adopted by Fugazza and Hoffmann (2017). Specifically, we added into the original gravity model (i.e., $\ln(\text{BTV}_{ij}) = \beta_0 + \beta_1 \times \ln(\text{GDP}_i \times \text{GDP}_j) + \beta_2 \times \ln(d_{ij}) + \varepsilon_{ij}$), the following three additional explanatory variables that are related with the GLSN, i.e., $\ln(\text{Gb}_i \times \text{Gb}_j)$, $\ln(\text{LSBCI}_{ij})$, and $\ln(\text{GLSNBC}_{ij})$. Then, we compared the performance between the multivariate regression models equipped with different explanatory variables. Results are shown in Table (a) below. First, our result is consistent with that by Fugazza and Hoffmann, i.e., LSBCI as an additional explanatory variable enhanced the explanatory power of the gravity model in the sense of AIC. Second, we found that our GLSN betweenness centrality (denoted as Gb) also enhanced the explanatory power of the gravity model, which is at a similar level as the LSBCI does. In the revised manuscript, we have described these results in a new paragraph with a table (lines 491-506, pages 14-15).

Note that the GLSNBC_{ij} in Table (a) denotes the number of edges between the ports of country i and country j in the GLSN, which is the quantity the reviewer proposed. However, the models and results using this variable are shown only in the present response letter, but not in the revised manuscript. This is because this variable is not used in the preceding analysis, and thus we prefer not to use it in the current analysis to keep the manuscript self-consistent.

Furthermore, when we used Gc (i.e., GLSN connectivity) instead of Gb , the results, shown below in Table (b), were similar as those for Gb . We have mentioned this in the revised manuscript (lines 505-506, page 15) and added Table (b) to the supplementary information.

Table (a). Multivariate regression results for gravity models in which the bilateral trade value of country pairs is the dependent variable, and the GLSN betweenness and LSBCI are additional explanatory variables. We considered 1773 country pairs, for which the two countries are directly connected in the GLSN (i.e., there exists at least one connection between the ports of two countries), the two countries' GDP values are available, the LSBCI value between the two countries is available, and the two countries' Gb values are positive. **: p-value < 0.001, *: p-value < 0.01, +: p-value < 0.05.

Explanatory variable	Adjusted R^2	AIC	Max VIF
$\ln(\text{GDP}_i \times \text{GDP}_j)$	0.683**	1803.06	1.00
$\ln(d_{ij})$	0.006**	3830.51	1.00
$\ln(\text{LSBCI}_{ij})$	0.348**	3083.64	1.00
$\ln(\text{Gb}_i \times \text{Gb}_j)$	0.290**	3233.26	1.00
$\ln(\text{GLSNBC}_{ij})$	0.368**	3026.81	1.00
$\ln(\text{GDP}_i \times \text{GDP}_j), \ln(d_{ij})$	0.778**	1175.23	1.23
$\ln(\text{GDP}_i \times \text{GDP}_j), \ln(\text{LSBCI}_{ij})$	0.686**	1785.80	1.77
$\ln(\text{GDP}_i \times \text{GDP}_j), \ln(\text{Gb}_i \times \text{Gb}_j)$	0.684**	1797.97	1.88

$\ln(\text{GDP}_i \times \text{GDP}_j), \ln(\text{GLSNBC}_{ij})$	0.720**	1585.44	1.41
$\ln(d_{ij}), \ln(\text{LSBCI}_{ij})$	0.364**	3040.63	1.13
$\ln(d_{ij}), \ln(\text{Gb}_i \times \text{Gb}_j)$	0.361**	3048.49	1.45
$\ln(d_{ij}), \ln(\text{GLSNBC}_{ij})$	0.377**	3002.95	1.00
$\ln(\text{LSBCI}_{ij}), \ln(\text{Gb}_i \times \text{Gb}_j)$	0.358**	3057.24	2.96
$\ln(\text{LSBCI}_{ij}), \ln(\text{GLSNBC}_{ij})$	0.490**	2648.61	1.27
$\ln(\text{Gb}_i \times \text{Gb}_j), \ln(\text{GLSNBC}_{ij})$	0.473**	2706.77	1.19
$\ln(\text{GDP}_i \times \text{GDP}_j), \ln(d_{ij}), \ln(\text{LSBCI}_{ij})$	0.785**	1120.65	1.94
$\ln(\text{GDP}_i \times \text{GDP}_j), \ln(d_{ij}), \ln(\text{Gb}_i \times \text{Gb}_j)$	0.787**	1102.86	2.23
$\ln(\text{GDP}_i \times \text{GDP}_j), \ln(d_{ij}), \ln(\text{GLSNBC}_{ij})$	0.786**	1106.58	1.96
$\ln(\text{GDP}_i \times \text{GDP}_j), \ln(\text{LSBCI}_{ij}), \ln(\text{Gb}_i \times \text{Gb}_j)$	0.696**	1731.66	3.33
$\ln(\text{GDP}_i \times \text{GDP}_j), \ln(\text{LSBCI}_{ij}), \ln(\text{GLSNBC}_{ij})$	0.721**	1582.91	2.02
$\ln(\text{GDP}_i \times \text{GDP}_j), \ln(\text{Gb}_i \times \text{Gb}_j), \ln(\text{GLSNBC}_{ij})$	0.722**	1574.46	2.23
$\ln(d_{ij}), \ln(\text{LSBCI}_{ij}), \ln(\text{Gb}_i \times \text{Gb}_j)$	0.402**	2932.28	4.03
$\ln(d_{ij}), \ln(\text{LSBCI}_{ij}), \ln(\text{GLSNBC}_{ij})$	0.492**	2643.00	1.51
$\ln(d_{ij}), \ln(\text{Gb}_i \times \text{Gb}_j), \ln(\text{GLSNBC}_{ij})$	0.491**	2646.90	1.93
$\ln(\text{LSBCI}_{ij}), \ln(\text{Gb}_i \times \text{Gb}_j), \ln(\text{GLSNBC}_{ij})$	0.497**	2625.39	3.17
$\ln(\text{GDP}_i \times \text{GDP}_j), \ln(d_{ij}), \ln(\text{LSBCI}_{ij}), \ln(\text{Gb}_i \times \text{Gb}_j)$	0.787**	1099.83	4.24
$\ln(\text{GDP}_i \times \text{GDP}_j), \ln(d_{ij}), \ln(\text{LSBCI}_{ij}), \ln(\text{GLSNBC}_{ij})$	0.791**	1072.45	2.42
$\ln(\text{GDP}_i \times \text{GDP}_j), \ln(d_{ij}), \ln(\text{Gb}_i \times \text{Gb}_j), \ln(\text{GLSNBC}_{ij})$	0.792**	1058.27	2.37
$\ln(\text{GDP}_i \times \text{GDP}_j), \ln(\text{LSBCI}_{ij}), \ln(\text{Gb}_i \times \text{Gb}_j), \ln(\text{GLSNBC}_{ij})$	0.727**	1539.59	3.35
$\ln(d_{ij}), \ln(\text{LSBCI}_{ij}), \ln(\text{Gb}_i \times \text{Gb}_j), \ln(\text{GLSNBC}_{ij})$	0.509**	2584.30	4.18
$\ln(\text{GDP}_i \times \text{GDP}_j), \ln(d_{ij}), \ln(\text{LSBCI}_{ij}), \ln(\text{Gb}_i \times \text{Gb}_j), \ln(\text{GLSNBC}_{ij})$	0.792**	1057.38	4.29

Table (b). Multivariate regression results for gravity models in which the bilateral trade value of country pairs is the dependent variable, and the GLSN connectivity and LSBCI are two additional explanatory variables. We considered 1818 country pairs, for which the two countries are directly connected in the GLSN (i.e., there exists at least one connection between the ports of two countries), the two countries' GDP values are available, and the LSBCI value between the two countries is available. **: p-value < 0.001, *: p-value < 0.01, +: p-value < 0.05.

Explanatory variable	Adjusted R^2	AIC	Max VIF
$\ln(\text{GDP}_i \times \text{GDP}_j)$	0.692**	1852.47	1.00
$\ln(d_{ij})$	0.008**	3979.67	1.00
$\ln(\text{LSBCI}_{ij})$	0.368**	3160.08	1.00
$\ln(\text{Gc}_i \times \text{Gc}_j)$	0.478**	2814.03	1.00
$\ln(\text{GLSNBC}_{ij})$	0.359**	3185.25	1.00
$\ln(\text{GDP}_i \times \text{GDP}_j), \ln(d_{ij})$	0.782**	1229.19	1.24
$\ln(\text{GDP}_i \times \text{GDP}_j), \ln(\text{LSBCI}_{ij})$	0.695**	1834.68	1.85
$\ln(\text{GDP}_i \times \text{GDP}_j), \ln(\text{Gc}_i \times \text{Gc}_j)$	0.694**	1845.39	3.72
$\ln(\text{GDP}_i \times \text{GDP}_j), \ln(\text{GLSNBC}_{ij})$	0.728**	1631.86	1.39
$\ln(d_{ij}), \ln(\text{LSBCI}_{ij})$	0.382**	3120.85	1.13

$\ln(d_{ij}), \ln(Gc_i \times Gc_j)$	0.595**	2352.36	1.43
$\ln(d_{ij}), \ln(GLSNBC_{ij})$	0.370**	3156.25	1.00
$\ln(LSBCI_{ij}), \ln(Gc_i \times Gc_j)$	0.488**	2778.12	2.61
$\ln(LSBCI_{ij}), \ln(GLSNBC_{ij})$	0.500**	2735.91	1.26
$\ln(Gc_i \times Gc_j), \ln(GLSNBC_{ij})$	0.530**	2624.48	1.57
$\ln(GDP_i \times GDP_j), \ln(d_{ij}), \ln(LSBCI_{ij})$	0.788**	1180.62	2.04
$\ln(GDP_i \times GDP_j), \ln(d_{ij}), \ln(Gc_i \times Gc_j)$	0.788**	1176.65	4.35
$\ln(GDP_i \times GDP_j), \ln(d_{ij}), \ln(GLSNBC_{ij})$	0.791**	1151.83	1.92
$\ln(GDP_i \times GDP_j), \ln(LSBCI_{ij}), \ln(Gc_i \times Gc_j)$	0.702**	1794.81	5.24
$\ln(GDP_i \times GDP_j), \ln(LSBCI_{ij}), \ln(GLSNBC_{ij})$	0.728**	1628.26	2.09
$\ln(GDP_i \times GDP_j), \ln(Gc_i \times Gc_j), \ln(GLSNBC_{ij})$	0.740**	1549.58	4.23
$\ln(d_{ij}), \ln(LSBCI_{ij}), \ln(Gc_i \times Gc_j)$	0.597**	2345.99	3.42
$\ln(d_{ij}), \ln(LSBCI_{ij}), \ln(GLSNBC_{ij})$	0.502**	2730.28	1.49
$\ln(d_{ij}), \ln(Gc_i \times Gc_j), \ln(GLSNBC_{ij})$	0.598**	2339.62	3.11
$\ln(LSBCI_{ij}), \ln(Gc_i \times Gc_j), \ln(GLSNBC_{ij})$	0.542**	2576.61	3.26
$\ln(GDP_i \times GDP_j), \ln(d_{ij}), \ln(LSBCI_{ij}), \ln(Gc_i \times Gc_j)$	0.789**	1165.29	6.29
$\ln(GDP_i \times GDP_j), \ln(d_{ij}), \ln(LSBCI_{ij}), \ln(GLSNBC_{ij})$	0.794**	1121.74	2.48
$\ln(GDP_i \times GDP_j), \ln(d_{ij}), \ln(Gc_i \times Gc_j), \ln(GLSNBC_{ij})$	0.792**	1143.79	6.09
$\ln(GDP_i \times GDP_j), \ln(LSBCI_{ij}), \ln(Gc_i \times Gc_j), \ln(GLSNBC_{ij})$	0.750**	1477.32	5.82
$\ln(d_{ij}), \ln(LSBCI_{ij}), \ln(Gc_i \times Gc_j), \ln(GLSNBC_{ij})$	0.601**	2328.92	5.78
$\ln(GDP_i \times GDP_j), \ln(d_{ij}), \ln(LSBCI_{ij}), \ln(Gc_i \times Gc_j), \ln(GLSNBC_{ij})$	0.794**	1123.73	8.73

Major corrections:

-Please have a native speaker check the manuscript for English grammar and structure of sentences.

Done accordingly. We have carefully checked the entire manuscript for English grammar and structure of sentences, and also have had qualified native speakers do so. We have uploaded to the journal's resubmission system a certificate of language-editing from a language editing service.

-P4.L32: The aim of the present study is mentioned, but there is not much information why this hypothesis is proposed and what evidence already exists to support this? This should be introduced much more clearly here. Also I would consider another term for 'middleman'

We have added the following text (lines 102-118, page 3), which introduces the relevant information on liner shipping industry practices and existent evidence to explain why our hypothesis is proposed.

“We hypothesize that the role of a port or country as broker to mediate liner shipping between different countries is correlated with the importance of the port or country in international trade. This hypothesis is consistent with both liner shipping industry practices and network theory. In liner shipping, a broker role may reflect the potential of a port/country to be a transshipment hub that facilitates container cargo transportation between other ports/countries. In fact, because the transshipment of containers has been a fastest growing segment of the container port market, container ports are fiercely competing for becoming transshipment hubs [28,29]. In network analysis, various centrality measures for nodes quantify the importance or role of nodes under the premise that the node's position impacts opportunities and constraints that it encounters [30,31]. In particular, the role as broker is often quantified by the betweenness centrality [32] or more succinctly by the degree (i.e., the number of edges that a node has). Nodes occupying structural holes may also benefit from the missing connections between their neighbors [33–35]. However, these or other centrality measures that neglect, for example, the nationality of the ports or the individual service routes may be poor indicators of countries' statuses in international trade and global economy.”

We have also replaced “middleman” by “broker” throughout the manuscript.

-P5.L10 The use of an undirected network may be flawed, in particular when looking at liner shipping where a lot of liner schedules have a pendulum network designs. I understand the choice of using an undirected network, but do at least reflect on this in this section or the discussion.

We agree. The original data do not allow us to obtain information about the directionality of the edges. Therefore, we added the following text to discuss this limitation in the Discussion section (lines 583-592, page 17).

“Another limitation of this study is that we have not considered the directionality of the edges in the GLSN. The information on the sequence of port calls in each service route is available in our data set. However, it remains challenging to infer the directionality of each inter-port connection in a service route, and hence the direction of edges between ports and between countries. This is because there may exist cargo transportation between two ports that are not sequentially called. For example, in a circular route that calls at ports A, B, C, D, E, C, and A in this order, any of the five ports may send cargos to any other port, rendering the estimation of direct edges difficult. Directed GLSNs, if one can reasonably estimate them, may contribute to improving the accuracy of describing import and export trade values.”

-It seems like the author have missed a major research article related to their work: Fugazza, M. and Hoffmann, J. (2017) ‘Liner shipping connectivity as determinant of trade’, *Journal of Shipping and Trade*. *Journal of Shipping and Trade*, 2(1), pp. 1–18. doi: 10.1186/s41072-017-0019-5.

Please make sure to include it and reflect on the differences in method and results.

Thanks for drawing our attention to this important reference. We cited it and added the following text (lines 537-541, page 16) to the Discussion section:

“A previous study has supported that improving bilateral connectivity in liner shipping can facilitate the bilateral trade between two countries, suggesting the influence of liner shipping connectivity on international trade at the country pair level [56]. The present study provides insights into understanding such influences at the country level, by focusing on the liner shipping connectivity of individual countries.”

-I am not convinced by the relationship found in 2.5. I don't see a clear explanation for the fact why the Gb2015 would be essential to explain growth, apart from the fact that trade is a global phenomenon and that a lot of countries see a similar increase in trade, hence the importance of Tv2015. I think the authors should think a bit deeper about the implications of setting up this model.

As the reviewer suggested, the importance of Tv2015 in explaining the growth is straightforward and natural. By contrast, the fact that Gb2015 is essential to explain the growth is what our regression results tell us. We set up this model because we expected that some of these GLSN-based explanatory variables are important in explaining the growth. This is our motivation to set up this model, and we explained the motivation in the first sentence of this section and also in various places in earlier sections where we emphasized the potential importance of our GLSN indicators. Although we cannot clarify mechanistic reasons why Gb2015 is essential to explain the growth, we believe that our discussion and statements on our modeling are sufficiently grounded.

It would be interesting to see if changes in the Gb2015 are an indicator of changes in trade, which could mean that if ports are becoming increasingly connected to other ports, it would explain the a larger increment in trade than a country where no changes in the Gb2015 were observed.

As we understand it, here the reviewer is probably suggesting that we analyze the correlation between “changes in Gb (= Gb2018 - Gb2015)” and “changes in trade value (= Tv2018 - Tv2015)”. In our study, we analyzed the correlation between “Gb2015” and “changes in trade value (= Tv2018 - Tv2015)”. We thank the reviewer for the comment but would prefer to keep the present analysis as it is, because it helps us test “whether a country's Gb value in a previous year may have a predictive power on the country's trade value change in following years” in a more straightforward way than the analysis that the reviewer suggested.

-Section 2.6. Instead of comparing the gravity model and GLSN, it would be more interesting to see what happens if the GLSN is added to the gravity equation: e.g.

can the trade between countries be better estimated by an improved maritime connectivity value?. I think this has much broader policy implications, as it means that the standard gravity equation, that relies on distance between countries as friction, is overly simplistic if these countries are not well connected (e.g. do not have a direct liner connection.).

We have carried out this analysis. Please see above for our reply, on pages 5-7 of this reply letter.

-The link to the GitHub does not yet exist. Please do create this before submitting for review.

Apology. We have created the content of the Github.

Minor corrections

-P3.L29: "important to" not good English. Change this sentence.

We changed it to "important for".

-P3.L29: with maritime countries, you mean countries that are not landlocked?

Yes. We have specified it in the main text as follows (line 60, page 2):

"Maritime countries (i.e., countries that are not landlocked) altogether account for"

-P3.L37: "as various proposed indices quantify". This is unclear, either change or remove.

We removed it.

-P3.L50: Sentence starting with "The LSCI" is very long, please break up.

We divided the sentence into two sentences (lines 74-79, page 2).

-P4.L13: 'even landlocked' is not good English.

We replaced "for even landlocked" by "even for landlocked".

-P4.L29: “and thus global economy”, should be “trade, and thus the global economy, but do”. Also this sentence is not entirely clear as it was just mentioned how the LSBCI is the connectivity between countries. I think what you are trying to say is the interconnectivity of all ports and countries?

To address these issues, we rewrote this sentence, entailing dividing it into two sentences. The revised text reads as follows (lines 99-101, page 3):

“These existing measures quantify how much individual countries or ports are integrated into international trade and the global economy. However, they do not tell how countries or ports are specifically connected to each other.”

-P5.L11: Why six different edge weighting methods? What are the weights of this network, the frequency, or the actual cargo trade?

From the method section this seems to be TEU. At least mention this briefly in the main text.

Yes. We have added text to the Results section to briefly mention this, which reads as follows (lines 292-294, page 8):

“We constructed one unweighted and six weighted undirected GLSNs (in terms of traffic capacity measured in TEU) from a data set of 1316 international liner shipping service routes in 2015.”

Please note that the six different edge weighting methods had been explained in detail in the subsection titled “Construction of the GLSN” (pages 4-5 in the current manuscript).

-P6.L53: I am not sure you need to justify keeping the LSCI in.

We had the following text in the previous version (lines 322-324, pages 9-10 in the present revised version) for the justification of keeping the LSCI.

“Fifth, we keep LSCI because it is an UNCTAD's official indicator, is the only one that we use and does not explicitly depend on our GLSN, and is reasonably strongly correlated with the trade value.”

To strengthen this, we have added the following text right after this sentence.

“Therefore, the LSCI serves as a benchmark indicator.”

-Results section: Would it be possible to make some maps with a few of the indicators and put them in the Supplement? This would help to give an idea of the differences and spatial distribution between the weight methods.

To the SI we have added four world maps, each of which shows the Gc, Gb, Fb, and LSCI values for the 157 countries (i.e., supplementary figure 1 in the SI). We have added text to refer to these figures in the main text (lines 325-326, page 10).

-L7.54: Please change “among the survivors” to “Among the remaining model configurations” or something similar.

We changed it as the reviewer proposed.

-P8.L12: The LSCI is excluded because of high collinearity? Or because it does not predict the results well enough?

We did not exclude the LSCI. It was simply not included in the two best models in terms of the AIC (see Table 2 in the main text, page 12). We explicitly explained this in the previous version (lines 338-358 on pages 10-11 in the present version, particularly lines 356-358).

-P8.L20: Great that you decided to do the same analysis for different trade metrics. Maybe it is good to reflect a bit on this; why the net export is less well correlated/explained by the model (which makes a lot of sense, but good to mention).

Thanks for noting this. We have added the following text (lines 386-391, pages 11-12):

“All the four explanatory variables ignore the information on the directionality of inter-port connections and that of international trades. Therefore, they are unable to distinguish the export and import value of a country, which the calculation of the net export of a country requires. We consider that this is a main reason why our regression models only marginally explain the net export value.”

-P9.L46: You don’t have to give a definition of how GDP is calculated.

We have removed the definition of how GDP is calculated.

-P9.L39: I am wondering whether it is relevant to explain GDP using the network indicators, as GDP is most likely indirectly correlated to the indicators through the trade value (as also included in the gravity equation). Or there is any other way in which the indicators are related to GDP, but I don’t see what this is. This also has a lot to do with the question whether or not growth in GDP leads to more investment

in port infrastructure and better connectivity, or if investment in port infrastructure lead to GDP growth. I would be careful making any claim about the relationship with GDP.

We thank the reviewer for the valuable comment. We have added the following text (lines 404-408, page 12) to briefly explain why we are interested in examining the relationship between GDP and the four GLSN-related variables.

“Previous studies on transport economics have shown that high performance of maritime transport logistics contributes to the economic growth of a country [49,50]. Therefore, in this section, we examine the extent to which the four explanatory variables, which are not direct derivatives of the GDP, are associated with the GDP.”

-P10.L29: Please rephrase the first sentence, I think you want to say “power on the changes in trade value over the years”.

We agree. We have rephrased this sentence as follows (lines 436-437, page 13):

“Next, we investigated whether the country’s position in the GLSN predicts changes in the trade value over time”.

-P13.L24 This last section is weak. I would like to see a concluding remark here, not a reference to other, non-relevant, literature. I think there is more relevant future work here related to the scope of the study.

We have replaced this original last section with the following concluding remark (lines 593-603, pages 17-18).

“In conclusion, we have provided evidence that countries’ positions in the GLSN are associated with their international trade status. Our results are expected to seed further research towards quantitatively understanding of the interplay between the structure of maritime shipping networks and international trade. For example, the global liner shipping system is evolving over time. Therefore, investigation of the dynamics of the GLSN and how they are related to dynamic changes in countries’ international trade status warrants future work. We also remark that liner shipping, though dominant in terms of the cargo value, is not the only mode of maritime shipping that facilitates the seaborne trade worldwide. Maritime shipping networks that incorporate multiple major shipping modes (e.g., liner shipping, dry bulk shipping, and oil shipping) [59,60] await further exploration.”

Reviewer 3

Dear authors,

Thank you for submitting this manuscript. I think the work is interesting, but I had to read it twice before I started to understand all the things described in this piece. Also, think about the message you want to convey with this paper. There are many results now on the multivariate regressions, yet the introduction seems to incline that it is going to be a network analysis. Hence, I am a bit lost what the main message is of the paper when reading the results. Please find below some comments in which I provide some feedback on the manuscript.

We are glad to hear an overall positive assessment and your constructive comments. We amended the manuscript as follows.

- In the introduction, you say: “The overarching goal of the present study is to apply network analysis on shipping networks to derive useful quantitative knowledge about international trade and its growth for individual countries”. Is that true? The results primarily focus on the results of the (multivariate) regression analysis, which is, to my knowledge, not a network analysis. Don’t you just extract some network characteristics from the GLSN and use those in the regression analysis?

We have accordingly replaced the sentence “the overarching goal of the present study” by the following one (lines 90-92, page 3):

“The aim of the present study is to evaluate the extent to which the information on the GLSN connectivity (i.e., which ports connect to which ports) helps us to estimate countries’ international trade status.”

With this aim, we proposed two GLSN indicators and also used some existing network and non-network indices (Freeman betweenness and the LSCI, respectively) to run multivariate linear regression. In our opinion, regression analyses whose variables include those derived from network analysis (i.e., GLSN indicators and Freeman betweenness) are network analysis. We believe that this is a common and reasonable understanding. Please note that this is already illustrated in the final paragraph of the Introduction section (lines 119-126, pages 3-4), which reads as follows:

“In the present study, we analyze port-level GLSNs, which we derived from a record of liner shipping services. We propose two GLSN-based indices for individual countries that quantify roles which each country plays as broker in international maritime transport. Although the two indices are analogous to the node's degree and betweenness, the new indices use the information on ports’ nationalities and on the individual service routes. Then, we show that the proposed indices account for the country’s international trade value fairly well. In particular, their performance, either alone or in combination, is better than that of the LSCI.”

- In the introduction, you say: "The LSBCI was found to have a significant impact on South Africa's bilateral trade flows with its trading partners". Maybe I don't understand the index, but how can an index have an impact on trade flows?

Thanks for pointing this out. We have revised this sentence as follows (lines 82-83, page 2).

"The LSBCI was found to be correlated with South Africa's bilateral trade values with its trading partners [11]."

- Coming back to the point of understandability, I think the manuscript is not really written in a way that a non-expert can easily get the gist of the article. For instance, I would encourage to move the methods from the end to before the results. I think RSOS does not have that structure, which may have been an artifact from a previous submission to a different journal? Without reading the methods first, it is very hard to understand the results. I would also move Figure 2 to the methods, it is not a result.

Thanks for great suggestions. We have moved the Methods section to before the Results section. We have also moved Figure 2 to the Methods section.

- Line 23-34 can be moved to the methods section

We moved the mentioned text to be merged with the local text in the Methods section. The revised text reads as follows (lines 194-197, page 6):

"In descriptive analysis and the multivariate linear regressions, we used the following four GLSN-related explanatory variables, each of which we measured for the individual countries. In particular, we examined the power of these four variables in explaining countries' trade values and their growth."

To accommodate this adjustment, we have swapped the order of the two subsections in the Methods section such that the subsection titled "Explanatory variables" comes before the subsection titled "Statistical models".

- What is the exact added value of the gravity model? The gravity model is only briefly explained in the methods section, and no words about the results of the gravity model in the discussion?

As we have carefully discussed in the section titled “Comparison with the gravity model” in the revised manuscript, the added value of the gravity model is that, it helps us evaluate if our GLSN indicators perform better than the standard gravity model in the following two senses.

First, as we had presented in the previous version already, we compared the performance of the gravity model and our GLSN-based linear regression, and the latter turned out to be better than the former. For details, please see lines 463-490 on page 14.

Second, in response to Reviewer #2, we have carried out an additional analysis to show that adding either the GLSN betweenness or GLSN connectivity value improves the gravity model in estimating the bilateral trade value. We described the new results in lines 491-506 on pages 14-15. For details, please see our detailed reply to Reviewer #2’s comment (pages 5-7 of this response letter). Following the present reviewer’s suggestion here, we have also mentioned the result of the gravity models in the Discussion section of the revised manuscript (lines 531-533, page 16), which reads as follows:

“Furthermore, we found that adding either GLSN betweenness or GLSN connectivity to the gravity model improved its ability in estimating the bilateral trade value between them.”

- I am not sure if RSOS requires a concluding section, but if so, please add one.

We checked the RSOS requirements to find that it does not require a concluding section. However, we have added a brief concluding remark (lines 593-603 on pages 17-18).

Overall, I think some of the main results are interesting and novel. I see the merit of this paper, but the way it’s written requires more work. It’s now primarily only easy to read for scholars in the maritime economics research field. As this is an interdisciplinary journal, I strongly suggest to rewrite it in such a way that the results are interesting and understandable to a more wider range of people.

Thanks for this comment. We tried to improve the readability throughout the manuscript by incorporating your comments and also the other reviewers’.

Appendix B

We thank the editor and the reviewers for evaluating our manuscript. We have further improved the manuscript in accordance with the second reviewer's valuable comments.

Response to Reviewer: 2

I am very satisfied with the way the authors have responded to my comments and the adjustments that are made. I still have a few style/grammar points to improve the manuscript. Additionally, I would like to emphasize that the main message that you re-iterate in the discussion: "These results support a long-standing view in maritime economics, which has yet been directly tested, that countries that are more strongly integrated into the global maritime transportation network have better access to global markets and thus greater trade opportunities" is really the main overall message and hypothesis here in my opinion. I think this is more important and clear (in terms of communication) compared to the main hypothesis in the article proposed now: "We hypothesize that the role of a port or country as broker to mediate liner shipping between different countries is correlated with the importance of the port or country in international trade." You can consider rewriting the article a little bit so that the first sentence is the main message, and the broker role is the means by which integration is facilitated in practise. I think in this way it could improve the communication of the article and its readability for a broader audience.

We are glad to hear that the reviewer is satisfied with our revision. We thank the reviewer for the valuable comment here, which we do agree with. We have modified the related text in the Introduction section as follows (lines 103-108, page 3):

"A long-standing premise in maritime economics is that countries that are strongly integrated into the global maritime transportation network have enhanced access to global markets and trade opportunities [9]. To operationally test this claim, we hypothesize that the role of a port or country as broker to mediate liner shipping between different countries is correlated with the importance of the port or country in international trade."

P2 L45: "Vessels travel from a port of one country to another on networks of ports to carry cargos, which to countries' international trade values." Please rewrite this sentence as it is unclear. Something in the order of: "Cargo loaded vessels travel from one country's port to another via an underlying port-to-port transport network."

Replaced by "Cargo-loaded vessels travel from one country's port to another via an underlying port-to-port transport network, contributing to international trade values of the countries en route."

P2L52: "international trade value" should be "bilateral trade flows."

We disagree. We believe that the present term “international trade value” is more accurate than the suggested term “bilateral trade flows” to describe what we study here. However, the reviewer’s comment led us to propose to further modify the related phrase (lines 52-53, page 2) as follows, for improving its readability to general readers:

“.....show that they explain a large amount of variation in individual countries’ international trade values.....”

P2L82: “bilateral trade values” and “trading partners” is double, please remove “with its trading partners” or write: “found to be correlated with SA’s trade with its trading partners.”

We removed “with its trading partners” as suggested by the reviewer.

P3L98: I don’t understand the word “orthogonal” in this sentence. Please change.

We replaced “is orthogonal to” by “is considered to be complementary to that provided by”.

P3L116: Please rephrase this sentence. First, I don’t understand what is meant with centrality measures that neglect the nationality of the ports or service routes. Second, why are these centrality measures poor indicators? I think I get what you are saying but this is quite a critical point for doing this research so I would like it emphasized more.

We modified this sentence (lines 118-122 in the present revised manuscript) as follows:

“However, these or other centrality measures largely use only the data about the network structure and neglect metadata, such as the nationality of the ports or the individual service routes. Such centrality measures based only on the network structure may be poor indicators of countries' statuses in international trade and global economy.”

P3L121: “quantify roles which each country plays as broker”, change to: “quantify each country’s role as broker in international maritime transport.”

Changed as suggested (lines 125-126 in the present revised manuscript).

P4L126: “better that of the LSCI.” I would phrase a bit more nuanced, such as: “is found to perform better than previously established liner connectivity metrics such as the LSCI.”

Thanks for a great suggestion. Changed as suggested (lines 130-131 in the present revised manuscript).

P4L131: “our previous studies”, to, “previous work.”

Changed as suggested (line 136 in the present revised manuscript).

P4L136: “The types of ship that data set covers” to ‘the vessel types included in the dataset are’

Changed as suggested (line 141 in the present revised manuscript), while we prefer to retain “data set”, which we believe is a more standard English wording.

P14L455: “These two variables, but not any other, were also”

Changed as suggested (line 460 in the present revised manuscript).

P15L515: Although it is great that you ran all combinations, a reader is only interested in the improvements compared to the original gravity model. Could you change the table to the original gravity model as baseline, and then followed by the extended models only. This makes it much easier to quickly see the improvements of it.

Thanks for a constructive comment. We have changed Table 3 as suggested by the reviewer. In the revised Table 3 (line 520 in the present revised manuscript), we retain only the original gravity model and the three extended models. We similarly modified Table S6 in the supplementary information.

P16L518: replace ‘stems’ with ‘originates’ or ‘is shaped by multiple’.

Replaced by ‘originates’ (line 523 in the present revised manuscript).

P16:519: replace ‘seek for profits’, ‘which primarily aim to maximise profits in a decentralized manner’.

Modified as requested (line 524 in the present revised manuscript).

P16L548: I would like to have this explanation of the structural hole theory in the introduction instead of discussion, because the structural hole theory was introduced there but not explained. It seems to be relevant for how you have constructed the centrality indicators, which is therefore essential information.

We thank the reviewer for the valuable comment. Instead of expanding on the structural hole theory in the Introduction section, we have removed the brief mentioning of the structural hole theory in the Introduction section, so that this issue

appears for the first time and is discussed in detail in the Discussion section. We believe that the remaining text in the Introduction section, together with the text in the Methods section, is sufficient for readers to understand why and how we construct our centrality indicators.

Response to Reviewer: 1

Comments to the Author(s)

The authors addressed all my comments.

We thank the reviewer once again for the valuable comments.